# Maximizing Intermediate Checkpoint Value in LLM Pretraining with Bayesian Optimization

**Deyuan Liu**[†][*][1] **Zecheng Wang**[†][*][1]

**Bingning Wang**[2] **Weipeng Chen**[3] **Chunshan Li**[1] **Zhiying Tu**[1] **Dianhui Chu**[1] **Dianbo Sui**[(✉)][1]

[†]deyuanliu@stu.hit.edu.cn    [†]zechengwang@stu.hit.edu.cn

## Abstract

The rapid proliferation of large language models (LLMs), such as GPT-4 and Gemini, underscores the intense demand for resources during their training processes, posing significant challenges due to substantial computational and environmental costs. In this paper, we introduce a novel checkpoint merging strategy aimed at making efficient use of intermediate checkpoints during LLM pretraining. This method utilizes intermediate checkpoints with shared training trajectories, and is rooted in an extensive search space exploration for the best merging weight via Bayesian optimization. Through various experiments, we demonstrate that: (1) Our proposed methodology exhibits the capacity to augment pretraining, presenting an opportunity akin to obtaining substantial benefits at minimal cost; (2) Our proposed methodology, despite requiring a given held-out dataset, still demonstrates robust generalization capabilities across diverse domains, a pivotal aspect in pretraining.

## 1. Introduction

With the rapid development of LLMs, such as GPT-3 (OpenAI, 2023), GPT-4 (OpenAI et al., 2023), PaLM (Chowdhery et al., 2023) and Gemini (Team et al., 2023), which boasts tens to hundreds of billions of parameters, the demand for new LLMs and the research aimed at enhancing their capabilities have significantly increased. But we should note that the training requirements for these LLMs are substantial, not only in terms of computational resources, human resources, and capital resources, but also regarding energy consumption and environmental impact. For

---
[*]Equal contribution [1]Harbin Institute of Technology [2]Tencent Wechat [3]Independent Researcher. Correspondence to: Dianbo Sui <suidianbo@hit.edu.cn>.

*Proceedings of the 42nd International Conference on Machine Learning*, Vancouver, Canada. PMLR 267, 2025. Copyright 2025 by the author(s).

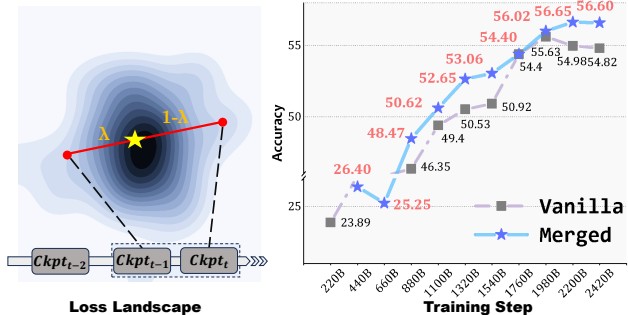

Figure 1: **Overview of the Bayesian optimization framework for checkpoint merging in LLM pretraining.** The framework operates by linearly combining intermediate checkpoints $\Theta_{t-1}$ and $\Theta_t$ with optimized merging weights $\lambda_t$. Through iterative Bayesian optimization, the method identifies performance "sweet spots" in the loss landscape that enhance model efficacy without much additional computational resources, effectively transforming intermediate checkpoints into improved models.

instance, training the LLaMA2 70B model with 2T tokens necessitates 1,720,320 GPU hours (Touvron et al., 2023), and the development of a transformer with 213 million parameters through neural architecture search can lead to environmental burdens equivalent to the lifetime $CO_2$ emissions of five cars over their entire lifespans (Strubell et al., 2019; Faiz et al., 2023). Consequently, making efficient use of checkpoints and the intermediate stages of the pretraining process has emerged as a key challenge in this field.

In response to this challenge, researchers have adopted various strategies in LLM pretraining, including mixed-precision training (Shoeybi et al., 2019), zero-redundancy optimizer (Rajbhandari et al., 2020), continuous retraining (Qin et al., 2022), pipeline parallelism (Liu et al., 2023), and depth up-scaling methods (Kim et al., 2023). Although these approaches contribute to efficient pretraining by optimizing model architecture and processes, they primarily focus on structural or optimization improvements rather than directly enhancing the utilization of intermediate checkpoints in the pretraining phase (Hou et al., 2022).

Unlike these studies, we focus on the model merging strategy, a classic topic in machine learning (Utans, 1996; Chen et al., 2017; Wortsman et al., 2022; Akiba et al., 2024; Li

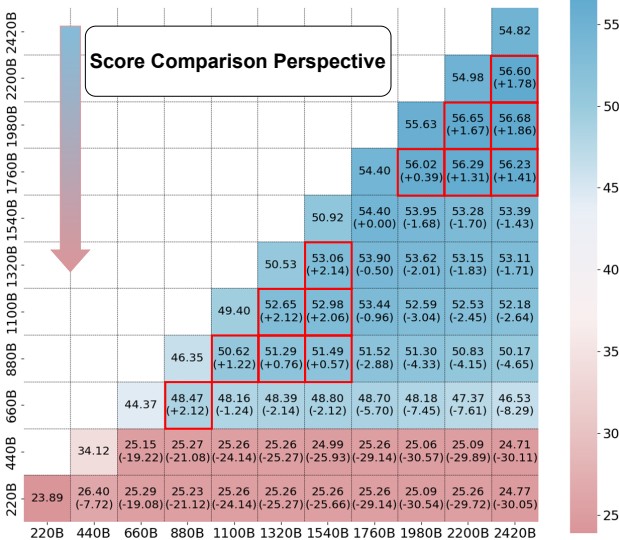

Figure 2: **Performance landscape of pairwise checkpoint merging using the Greedy Soup method on the C-Eval benchmark across 11 Baichuan2 checkpoints spanning 200B to 2640B tokens.** The heatmap reveals that merging adjacent checkpoints (near the diagonal) generally yields superior performance, while merging distant checkpoints results in significant performance degradation.

et al., 2023b; Yang et al., 2023b), to enhance LLM pretraining in this paper. In particular, we employ checkpoints saved during pretraining and average these checkpoint parameters to improve pretraining without requiring substantial resources, since merged checkpoints can reduce the variance of the combined output relative to the output of the individual checkpoint while not increasing the bias (Utans, 1996).

However, conducting checkpoint merging is not trivial in pretraining, because different local minima may be found in averaging parameters (Utans, 1996; Chen et al., 2017). Therefore it is important to investigate the basic characters of checkpoint merging and wisely determine the merging weight.

**Our Approach.** To this end, we make the following effort: (1) we conduct some pilot experiments to explore the characters of checkpoint merging; (2) Based on the findings in the pilot experiments, we propose a method rooted in Bayesian optimization to find the optimal or near-optimal merging weight. In detail, we first explore two research questions: *Which checkpoints in the pretraining trajectory should be merged?* and *How to merge checkpoint?* via various pilot experiments. Then, based on findings in pilot experiments, we leverage Bayesian optimization to optimize the expensive, black-box, and derivative-free objective function of checkpoint merging, and determine the checkpoint merging weight.

Through various experiments, we mainly find that:

(a) Our proposed approach has the potential to enhance

pretraining by efficiently utilizing intermediate checkpoints;

(b) Besides superior performance, the merged soup [1], determined by a specific held-out dataset same as Wortsman et al. (2022); Matena & Raffel (2022), still exhibits strong generalization capabilities across various domains, a crucial aspect in pretraining.

**Contributions:** In summary, the contribution of this paper is threefold:

(a) We propose merging checkpoints in the pretraining trajectory to make efficient use of checkpoints, offering substantial improvements without additional resource requirements.

(b) To find the optimal merging weight, we leverage Bayesian optimization, which excels at optimizing expensive black-box derivative-free objective functions.

(c) Through various experiments, we denote our method exhibits superior performance and the newly merged checkpoint maintains strong generalization across different domains.

## 2. Pilot Experiments

We conducted comprehensive pilot experiments to address two fundamental research questions that guide our checkpoint merging strategy:

> $\mathbb{RQ}1$: *Which checkpoints along the pretraining trajectory should be merged?*
>
> $\mathbb{RQ}2$: *How should these checkpoints be merged optimally?*

### 2.1. Experimental Setup

To systematically investigate these research questions, we selected eleven representative checkpoints from the Baichuan2 model (Yang et al., 2023a), spanning a comprehensive range from *200B* to *2640B* tokens during pretraining. We evaluated the merged checkpoints using C-Eval (Huang et al., 2023), a rigorous benchmark encompassing *52* subjects across four difficulty levels, providing comprehensive coverage of language understanding capabilities. All merging experiments employed the greedy soup strategy (Wortsman et al., 2022), where checkpoints are combined sequentially, with each checkpoint added only if it demonstrates measurable improvement in accuracy on a held-out development set.

### 2.2. $\mathbb{RQ}1$: Strategic Identification of Mergeable Checkpoints

To systematically address $\mathbb{RQ}1$, we conducted an exhaustive exploration of all pairwise combinations among the

---

[1]According to Wortsman et al. (2022), the merged result is called "soup".

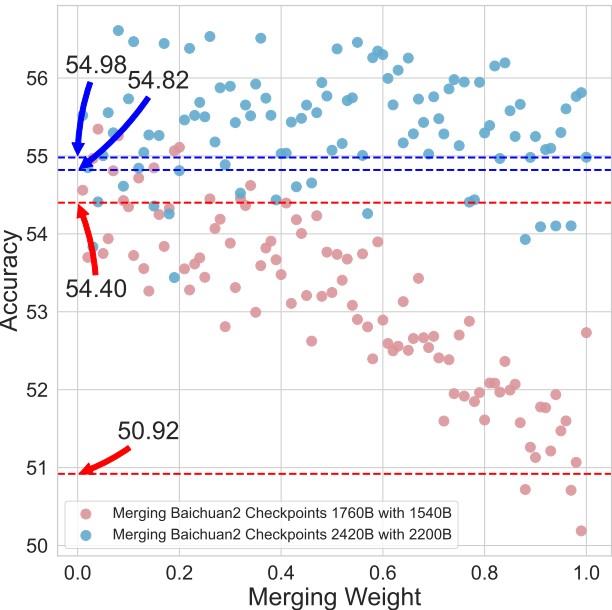

Figure 3: **Impact of varying merging weights on model performance when combining two representative checkpoint pairs:** Baichuan2-1540B with Baichuan2-1760B and Baichuan2-2200B with Baichuan2-2420B. The graph illustrates accuracy on the C-Eval dataset as a function of uniformly sampled merging weights ranging from 0 to 1. The results demonstrate distinct patterns: for checkpoints with performance gaps, optimal weights favor the stronger model, while for similarly performing checkpoints, a broad range of weights **(76%)** can yield improvements, highlighting the complexity of the optimization landscape.

eleven Baichuan2 checkpoints, yielding *55* distinct merging scenarios. The performance of these merged models was rigorously evaluated on the C-Eval test set, with results visualized in Figure 2 .

**Key Findings.** Our analysis reveals that merges involving **adjacent checkpoints** in the pretraining trajectory consistently led to substantial performance improvements compared to individual models. Most notably, merging Baichuan2-1980B with Baichuan2-2200B achieved an accuracy of **56.65%**, significantly outperforming Baichuan2-2200B alone (**54.98%**). Remarkably, this merged model surpassed even the final checkpoint, Baichuan2-2420B (**54.82%**), by a substantial margin of **1.83%**. These encouraging trends were consistently observed across the CMMLU benchmark (detailed analysis in Appendix J ).

Conversely, merging **distant checkpoints** often resulted in severe performance degradation. For instance, combining the early-stage Baichuan2-220B with the mature Baichuan2-2200B yielded an accuracy of merely **25.26%**, only marginally exceeding the undertrained Baichuan2-220B baseline (**23.89%**), indicating destructive interference between disparate training states.

## 2.3. $\mathbb{RQ}2$: Optimal Strategies for Merging Checkpoints

To explore $\mathbb{RQ}2$, we examined the impact of varying merging weights when combining two representative pairs of checkpoints: Baichuan2-1540B with Baichuan2-1760B, and Baichuan2-2200B with Baichuan2-2420B. For each pair, we uniformly sampled 100 weights from the interval [0, 1] and evaluated each merged model on C-Eval, as illustrated in Figure 7 .

**Results for Distant Checkpoints.** When merging checkpoints with a substantial performance gap, such as Baichuan2-1540B and Baichuan2-1760B, the accuracy increased smoothly as the weight on the stronger checkpoint (Baichuan2-1760B) increased. Notably, **13%** of the tested weights resulted in improvements beyond the stronger base model.

**Results for Similar Checkpoints.** In contrast, when merging two similarly strong checkpoints (Baichuan2-2200B and Baichuan2-2420B), the relationship between merging weight and performance did not follow a strict monotonic pattern. Instead, a broad range of weights **76%** led to performance improvements over the stronger checkpoint. A similar pattern was observed in experiments merging DeepSeek's 7B checkpoints (DeepSeek-AI et al., 2024) at 1800B and 2000B tokens (see Appendix F ).

**Summary** Our pilot experiments provide valuable insights into the strategies for merging model checkpoints. Specifically, merging adjacent checkpoints along the pretraining trajectory generally enhances performance, while merging distant checkpoints can be detrimental. Additionally, the choice of merging weights plays a crucial role, especially when combining checkpoints of similar strength, where a wide range of weights can yield performance gains. These findings inform our approach to checkpoint merging in subsequent methods.

**Additional Analyses** Further analyses on the CMMLU benchmark and DeepSeek's checkpoints are provided in Appendix J and Appendix F , respectively, demonstrating the generalizability of our findings across different evaluation metrics and model configurations.

## 3. A Bayesian Approach to Checkpoint Merging in LLM Pretraining

In this section, we present our novel framework for checkpoint merging during the pretraining of large language models (LLMs). We begin by formally defining the checkpoint merging process. Next, we introduce our optimized approach that leverages Bayesian optimization to determine the optimal merging weights. Finally, we explore the advantages of our method in generalization, with detailed proofs provided in Appendix A .

---

**Algorithm 1** Checkpoint Merging via Bayesian Optimization

---

1: **Input:** Initial checkpoints $\Theta_{t-1}$, $\Theta_t$, validation dataset $\mathcal{D}$, search bounds $[\alpha, 1]$, number of iterations $N$
2: Evaluate initial merging weights $\lambda_t^{(i)}$ (e.g., $\lambda_t^{(1)} = \alpha$, $\lambda_t^{(2)} = 1$) and collect observations $\mathcal{O} = \{(\lambda_t^{(i)}, f(\lambda_t^{(i)}))\}_{i=1}^{k_0}$
3: **for** $k = k_0 + 1$ to $N$ **do**
4:      Fit a Gaussian Process (GP) to the current observations $\mathcal{O}$
5:      Select the next merging weight $\lambda_t^{(k)} = \arg\max_{\lambda_t \in [\alpha, 1]} A(\lambda_t)$ using the acquisition function $A$
6:      Merge checkpoints: $\widetilde{\Theta}_t^{(k)} = \lambda_t^{(k)} \Theta_t + (1 - \lambda_t^{(k)}) \Theta_{t-1}$
7:      Evaluate the performance $f(\lambda_t^{(k)})$ of $\widetilde{\Theta}_t^{(k)}$ on the validation dataset $\mathcal{D}$
8:      Update the observations: $\mathcal{O} = \mathcal{O} \cup \{(\lambda_t^{(k)}, f(\lambda_t^{(k)}))\}$
9: **end for**
10: **Output:** Optimal merging weight $\lambda_t^* = \arg\max_{\lambda_t} \{f(\lambda_t) \mid (\lambda_t, f(\lambda_t)) \in \mathcal{O}\}$

---

### 3.1. Checkpoint Merging

During the pretraining of LLMs, the model periodically saves checkpoints denoted as $\{\Theta_1, \Theta_2, \ldots, \Theta_t\}$, representing the model parameters at various training iterations. A straightforward strategy to enhance model performance involves linearly combining these checkpoints in the parameter space, a technique commonly referred to as *Checkpoint Soup*. This can be mathematically expressed as:

$$\widetilde{\Theta}_t = \sum_{i=1}^{t} \lambda_i \Theta_i \quad \textbf{s.t.} \quad \sum_{i=1}^{t} \lambda_i = 1 \tag{1}$$

where $\lambda_i$ are the merging weights assigned to each checkpoint.

However, as the number of checkpoints increases, efficiently utilizing them becomes increasingly challenging due to the high-dimensional optimization problem that emerges from the growing number of weighting coefficients $\{\lambda_i\}$. In the general scenario, merging $t$ checkpoints necessitates the optimization of $t$ weights $\{\lambda_1, \lambda_2, \ldots, \lambda_t\}$ subject to the constraint $\sum_{i=1}^{t} \lambda_i = 1$. This constitutes a $(t-1)$-dimensional optimization problem, which rapidly escalates in complexity as $t$ grows. The exponential expansion of the search space renders exhaustive search and grid search methods computationally impractical for large $t$. And we empirically validate the efficacy of the pairwise merging strategy, which show in Appendix 5.3 .

To address this, we adopt a *pairwise merging* strategy, which significantly simplifies the optimization landscape by restricting the merging process to only the two most recent checkpoints at each step. This approach effectively reduces the problem to a one-dimensional search over the merging weight $\lambda_t$, thereby enhancing computational efficiency and scalability. Formally, the pairwise merging process is defined as:

$$\widetilde{\Theta}_t = \lambda_t \Theta_t + (1 - \lambda_t) \Theta_{t-1} \tag{2}$$

where $\lambda_t \in [\alpha, 1]$ and $\alpha \in (0, 1)$ serves as a lower bound to constrain the search space.

Empirical evidence indicates that the merged checkpoint $\widetilde{\Theta}_t$ can outperform the most recent checkpoint $\Theta_t$ when an optimal or near-optimal $\lambda_t$ is selected. The primary challenge lies in accurately determining this optimal merging weight $\lambda_t$, which we address by leveraging Bayesian optimization to maximize the effective utilization of intermediate checkpoints in the subsequent subsection.

Furthermore, our theoretical analysis ( Appendix A ) provides insights into why checkpoint merging can enhance model performance and convergence. Under realistic assumptions about the neural network loss landscape (such as smoothness, quadratic approximation near local minima, and bounded Hessians), we derive tighter bounds on the performance of the merged model $\widetilde{\Theta}_t$. Specifically, we establish that:

$$f(\widetilde{\Theta}_t) \approx \lambda_t f(\Theta_t) + (1 - \lambda_t) f(\Theta_{t-1}) \pm \Delta_t \tag{3}$$

where $f(\Theta)$ denotes the expected performance metric (e.g., accuracy), and $\Delta_t$ captures higher-order terms related to the curvature of the loss landscape and the distance between checkpoints.

Besides, our convergence analysis ( Appendix B ) demonstrates that checkpoint merging can influence the convergence behavior of gradient-based optimization algorithms. By interpolating between consecutive checkpoints, merging can effectively perform larger steps in parameter space, potentially accelerating convergence under certain conditions. Additionally, merging can smooth out fluctuations in the optimization trajectory caused by stochastic gradients, leading to more stable convergence.

### 3.2. Bayesian Optimization for Determining Merging Weights

Accurately determining the optimal merging weight $\lambda_t$ that maximizes the model's performance is the central challenge in checkpoint merging. To address this, we employ Bayesian optimization, an effective global optimization strategy for functions that are expensive to evaluate or lack closed-form expressions (Frazier, 2018).

**Optimization Objective** We formulate the optimization problem as:

$$\lambda_t^* = \arg \max_{\lambda_t \in [\alpha, 1]} f\big(\widetilde{\Theta}_t(\lambda_t)\big) \tag{4}$$

where $f\big(\widetilde{\Theta}_t(\lambda_t)\big)$ represents the performance of the merged model on a validation dataset as a function of the merging weight $\lambda_t$. The performance metric could be accuracy, perplexity, or any relevant evaluation metric for LLMs.

**Search Space Constraint** By setting $\lambda_t \in [\alpha, 1]$, we constrain the search space to merging weights that retain a significant contribution from the latest checkpoint $\Theta_t$. This aligns with practical observations that heavily weighting $\Theta_t$ often leads to better performance, as it captures the most recent training progress.

**Gaussian Process Regression** We model the objective function $f(\lambda_t)$ using Gaussian process (GP) regression (Rasmussen, 2003), which provides a probabilistic framework for modeling unknown functions. The GP defines a prior over functions, characterized by a mean function $\mu_0(\lambda_t)$ and a covariance function $k(\lambda_t, \lambda_t')$. We assume:

$$f(\lambda_t) \sim \mathcal{GP}\left(\mu_0(\lambda_t), k(\lambda_t, \lambda_t')\right) \tag{5}$$

where $\mu_0(\lambda_t)$ is the prior mean (often set to zero), and $k(\lambda_t, \lambda_t')$ is the kernel function encoding assumptions about the smoothness of $f(\lambda_t)$.

We collect a set of observations $\mathcal{D}_{\text{obs}} = \{(\lambda_t^{(i)}, f^{(i)})\}_{i=1}^N$ by evaluating the performance at $N$ different merging weights. The GP posterior mean $\mu_N(\lambda_t)$ and variance $\sigma_N^2(\lambda_t)$ are updated based on these observations:

$$\mu_N(\lambda_t) = \mu_0(\lambda_t) + k^\top(\lambda_t) K^{-1}(\mathbf{f} - \boldsymbol{\mu}_0), \tag{6}$$

$$\sigma_N^2(\lambda_t) = k(\lambda_t, \lambda_t) - k^\top(\lambda_t) K^{-1} k(\lambda_t) \tag{7}$$

where $k(\lambda_t) = [k(\lambda_t, \lambda_t^{(1)}), \ldots, k(\lambda_t, \lambda_t^{(N)})]^\top$, $K$ is the covariance matrix of the observations with elements $K_{ij} = k(\lambda_t^{(i)}, \lambda_t^{(j)})$, and $\mathbf{f}$, $\boldsymbol{\mu}_0$ are vectors of observed performance values and prior means, respectively.

Our theoretical analysis ( Appendix C ) demonstrates that using GP-based Bayesian optimization can efficiently converge to the optimal merging weight $\lambda_t^*$. By capturing the function's behavior, including any non-convexities or multi-modalities, GPs enable the discovery of $\lambda_t$ values that yield better performance than those found via grid or random search.

**Acquisition Function Selection** The acquisition function determines the next merging weight $\lambda_t$ to evaluate by balancing exploration of the search space and exploitation of known high-performing regions. We consider three acquisition functions: Expected Improvement (EI), Probability

of Improvement (PI), and Upper Confidence Bound (UCB). These are formally defined as:

$$A_i(\lambda_t) = \begin{cases} \mathbb{E}_k\left[\max(f(\lambda_t) - f_*^k, 0)\right], & \text{for EI} \\ \mathbb{P}(f(\lambda_t) > f_*^k), & \text{for PI} \\ \mu_k(\lambda_t) + \beta \cdot \sigma_k(\lambda_t), & \text{for UCB} \end{cases} \tag{8}$$

where $A_i(\lambda_t)$ represents the acquisition function used to select the next weight $\lambda_t^{(k+1)}$, $f_*^k$ is the best observed performance up to iteration $k$, $\mu_k$ and $\sigma_k$ denote the mean and standard deviation of the posterior distribution at $\lambda_t$, respectively, and $\beta$ is a tunable parameter controlling the exploration-exploitation trade-off.

To dynamically select the most promising acquisition function, we employ the GP-Hedge strategy, which combines multiple acquisition functions based on their past performance. At each iteration $k$, GP-Hedge updates the cumulative reward $R_i(k)$ for each acquisition function $i$:

$$A_{\text{GP-Hedge}}(\lambda_t) = \sum_{i=1}^M \frac{\exp(\eta R_i(k))}{\sum_{j=1}^M \exp(\eta R_j(k))} \cdot A_i(\lambda_t) \tag{9}$$

where $M$ is the number of acquisition functions (here, $M = 3$), $A_i(\lambda_t)$ represents the individual acquisition functions (EI, PI, or UCB), $R_i(k)$ is the cumulative reward for acquisition function $i$ at iteration $k$, and $\eta$ controls the learning rate. This strategy allows the acquisition function to adapt dynamically, leveraging the strengths of each component based on their historical performance.

The overall procedure is summarized in Algorithm 1 . We begin with initial observations (e.g., evaluating $\lambda_t = \alpha$ and $\lambda_t = 1$) and iteratively select new merging weights using Bayesian optimization until convergence or a predefined budget of evaluations.

### 3.3. Impact on Model Generalization

Checkpoint merging aims not only to improve performance on the validation set but also to enhance generalization to unseen data. Our analysis ( Appendix D ) employs the PAC-Bayesian framework to derive a generalization bound for the merged model $\widetilde{\Theta}_t$.

**PAC-Bayesian Generalization Bound** We define the prior distribution $P$ and posterior distribution $Q$ over the model parameters as follows:

$$P = \mathcal{N}\left(\Theta_{t-1}, \sigma_P^2 I\right), Q = \mathcal{N}\left(\widetilde{\Theta}_t, \sigma_P^2 I\right) \tag{10}$$

where $\mathcal{N}(\mu, \Sigma)$ denotes a Gaussian distribution, and $I$ is the identity matrix. The KL divergence between $Q$ and $P$ is:

$$D_{\text{KL}}(Q \parallel P) = \frac{\lambda_t^2}{2\sigma_P^2} \|\Theta_t - \Theta_{t-1}\|^2 \tag{11}$$

Table 1: **Comprehensive performance comparison of different checkpoint merging methods applied to Baichuan2 and DeepSeek models across multiple benchmark datasets.** The table evaluates four checkpoint pairs using various merging strategies: Uniform Soup, Greedy Soup, Fisher Weighted Averaging, RegMean, and our proposed Bayesian Optimization-based method. Underlined scores indicate the highest performance achieved by baseline methods within each model pair. Red highlighted improvements demonstrate our method's consistent superiority, achieving average improvements across different model pairs and maintaining robust performance across diverse evaluation metrics.

| Dataset | Baichuan2-1980B | Baichuan2-2200B | Uniform Soup | Greedy Soup | Fisher | RegMean | Ours |
|---|---|---|---|---|---|---|---|
| C-Eval(5-shot) | 55.63 | 54.98 | 53.00 | 55.63 | 55.73 | 55.21 | **56.17**(**+0.44**) |
| CMMLU(5-shot) | 55.68 | 56.29 | 54.20 | 56.29 | 56.13 | 55.21 | **56.88**(**+0.59**) |
| MMLU(5-shot) | 54.00 | 51.27 | 54.30 | 55.39 | 54.25 | 54.77 | **55.44**(**+0.05**) |
| GSM8K(4-shot) | 23.28 | 21.99 | 23.96 | 23.28 | 20.92 | 23.73 | **24.02**(**+0.29**) |
| Average | 47.15 | 46.13 | 46.37 | 47.65 | 46.76 | 47.23 | **48.13**(**+0.48**) |

| Dataset | Baichuan2-2200B | Baichuan2-2420B | Uniform Soup | Greedy Soup | Fisher | RegMean | Ours |
|---|---|---|---|---|---|---|---|
| C-Eval(5-shot) | 54.98 | 54.82 | 54.93 | 55.64 | 54.44 | 54.55 | **56.23**(**+0.59**) |
| CMMLU(5-shot) | 56.29 | 56.78 | 56.71 | 56.78 | 56.62 | 56.46 | **56.97**(**+0.19**) |
| MMLU(5-shot) | 51.27 | 53.97 | 54.62 | **54.82** | 54.16 | 54.77 | 54.56(-0.26) |
| GSM8K(4-shot) | 19.64 | 21.00 | 20.92 | 21.92 | 22.44 | 23.88 | **24.32**(**+0.44**) |
| Average | 45.55 | 46.64 | 46.80 | 47.29 | 46.92 | 47.42 | **48.02**(**+0.50**) |

| Dataset | DeepSeek-1400B | DeepSeek-1600B | Uniform Soup | Greedy Soup | Fisher | RegMean | Ours |
|---|---|---|---|---|---|---|---|
| C-Eval(5-shot) | 38.80 | 39.40 | 41.26 | 40.70 | 40.24 | 39.55 | **41.79**(**+0.55**) |
| CMMLU(5-shot) | 40.27 | 40.94 | 42.18 | 42.25 | 41.76 | 41.80 | **42.55**(**+0.30**) |
| MMLU(5-shot) | 41.94 | 42.60 | 43.87 | **43.88** | 43.95 | 43.27 | 43.85(-0.03) |
| GSM8K(4-shot) | 11.30 | 13.27 | 14.18 | 14.03 | 15.39 | 15.04 | **15.70**(**+0.41**) |
| Average | 33.08 | 34.05 | 35.37 | 35.22 | 35.34 | 34.92 | **35.97**(**+0.53**) |

| Dataset | DeepSeek-1800B | DeepSeek-2000B | Uniform Soup | Greedy Soup | Fisher | RegMean | Ours |
|---|---|---|---|---|---|---|---|
| C-Eval(5-shot) | 43.05 | 44.36 | 44.61 | 44.70 | 44.81 | 43.95 | **45.82**(**+1.01**) |
| CMMLU(5-shot) | 45.31 | 46.82 | 46.84 | 46.82 | 46.49 | 47.12 | **47.15**(**+0.03**) |
| MMLU(5-shot) | 47.68 | 49.29 | 49.02 | 49.29 | 48.73 | 49.07 | **49.43**(**+0.14**) |
| GSM8K(4-shot) | 16.60 | 18.88 | 17.82 | 18.88 | 18.73 | 18.56 | **19.04**(**+0.22**) |
| Average | 38.16 | 39.84 | 39.57 | 39.92 | 39.69 | 39.68 | **40.36**(**+0.44**) |

Using the PAC-Bayesian generalization bound (McAllester, 1998), it holds with probability at least $1 - \delta$:

$$\mathbb{E}_{\Theta \sim Q}\left[\mathcal{L}_{\mathcal{D}}(\Theta)\right] \leq \mathcal{L}_S(Q) + \sqrt{\frac{D_{\mathrm{KL}}(Q \parallel P) + \ln\left(\frac{2\sqrt{n}}{\delta}\right)}{2n}} \tag{12}$$

where $\mathcal{L}_{\mathcal{D}}(\Theta)$ is the expected loss on the data distribution, $\mathcal{L}_S(Q)$ is the empirical loss on the training set, and $n$ is the number of training samples.

Since $\lambda_t \leq 1$, the KL divergence in (Eq. (11)) is reduced compared to using $\Theta_t$ alone. This leads to a tighter generalization bound in (Eq. (12)), indicating that checkpoint merging can improve generalization by effectively regularizing the model.

## 4. Experiments

### Setup

• *Datasets & LLM Checkpoints.* We evaluate our method using multiple pretraining checkpoints and a broad range of datasets. For models, we follow previous Baichuan2 (Yang et al., 2023a) 7B, DeepSeek (DeepSeek-AI et al., 2024) 7B and Pythia (Biderman et al., 2023), ranging from 70M to 6.9B parameters models. For benchmarks, we evaluate on C-Eval (Huang et al., 2023), CMMLU (Li et al., 2023a),

MMLU (Hendrycks et al., 2020), and GSM8K (Cobbe et al., 2021), PIQA (Bisk et al., 2020), WinoGrand (Sakaguchi et al., 2021), SciQ (Welbl et al., 2017), and ARC-Easy (Clark et al., 2018).

• *Baseline Merging Methods.* We compare against strong baseline merging methods, including Uniform Soup and Greedy Soup (Wortsman et al., 2022), Fisher Weighted Averaging (Matena & Raffel, 2022), and RegMean (Jin et al., 2022). Uniform Soup evenly averages model parameters. Greedy Soup incrementally adds checkpoints that improve performance on a held-out set.

### 4.1. Main Results

Table 1 presents the comprehensive results of our merging experiments, highlighting two strategically selected checkpoint combinations that yield substantial performance enhancements. By merging the Baichuan2-1980B and Baichuan2-2200B checkpoints, our method achieves a remarkable **0.59%** improvement on the CMMLU dataset compared to the Baichuan2-2200B checkpoint alone. Similarly, merging the Baichuan2-2200B and Baichuan2-2420B checkpoints results in a significant **0.59%** improvement on the C-Eval dataset compared to the Baichuan2-2420B checkpoint. These results underscore the efficacy of our approach in enhancing model performance during the mid to late stages of pretraining.

**Comparison with Baseline Methods.** Beyond surpassing individual baseline checkpoints, our method significantly outperforms existing merging baselines across diverse datasets. Specifically, on the CMMLU dataset, merging the Baichuan2-1980B and Baichuan2-2200B checkpoints using our approach leads to improvements of **2.68%**, **0.59%**, **0.75%**, and **1.67%** over Uniform Soup, Greedy Soup, Fisher Weighted Averaging, and RegMean, respectively. A consistent trend is observed on the C-Eval dataset, where our method outperforms the baselines by substantial margins, demonstrating superior effectiveness in checkpoint merging optimization.

**Cross-Architecture Generalizability.** To validate the robustness and broad applicability of our proposed method, we applied it to the DeepSeek 7B architecture. The results, presented in Table 1, demonstrate that our method consistently enhances model performance across different pretraining stages and model architectures. This highlights the universal generalizability of our approach beyond the Baichuan2 family, establishing its effectiveness as a model-agnostic optimization strategy.

### 4.2. Generalization to Unseen Domains

To rigorously assess the generalization capabilities of merged checkpoints to unseen domains, we evaluated the out-of-domain (OOD) performance of various merging methods. Although merging weights were determined using the C-Eval dataset (in-domain, IND), we tested the merged models on three diverse OOD datasets: CMMLU, MMLU, and GSM8K.

Table 2 summarizes the results, revealing two critical insights:

(a) **Cross-Lingual Generalization:** Despite determining merging weights using a Chinese dataset (C-Eval), the merged checkpoints consistently perform well on English datasets such as MMLU and GSM8K. This indicates that merging pretraining checkpoints in parameter space preserves the models' ability to generalize across languages effectively.

(b) **Superior Stability and Cross-Domain Performance:** Our proposed method outperforms Greedy Soup and Fisher Weighted Averaging by exhibiting the smallest average absolute difference ($\Delta$) between IND and OOD performance. Specifically, our method achieves a $\Delta$ of **0.87**, compared to **2.01** and **2.38** for Greedy Soup and Fisher Weighted Averaging, respectively. This demonstrates that our Bayesian optimization approach identifies merging weights that are more optimal across different domains.

### 4.3. Performance Across Different Model Sizes

To evaluate whether our proposed merging method maintains effectiveness across LLMs with varying parameter sizes, we conducted comprehensive experiments on Pythia

Table 2: **Out-of-domain generalization analysis of merged checkpoint soups across language and task boundaries.** Merging weights are optimized using the C-Eval dataset (Chinese, in-domain), then evaluated on out-of-domain datasets: CMMLU, MMLU, and GSM8K. The IND/OOD format shows in-domain versus out-of-domain performance. The $\Delta$ metric quantifies the total absolute performance difference across domains, where lower values indicate superior generalization stability. Our method achieves the smallest $\Delta$ (**0.87**), demonstrating robust cross-lingual and cross-task generalization capabilities compared to baseline approaches.

| Dataset | Greedy Soup (IND/OOD) | Fisher (IND/OOD) | Ours (IND/OOD) |
|---|---|---|---|
| CMMLU | 56.78/56.78 | 56.62/56.72 | 56.97/56.91 |
| MMLU | 54.82/54.54 | 54.16/54.54 | 54.56/55.29 |
| GSM8K | 21.92/23.65 | 22.44/24.34 | 24.32/24.40 |
| $\Delta(\downarrow)$ | 2.01 | 2.38 | **0.87** |

models ranging from *70M* to *6.9B* parameters. We assessed performance on PIQA, WinoGrande, SciQ, and ARC-Easy datasets, with detailed PIQA results presented in Table 7.

**Key Findings.** Across all parameter sizes, our method consistently outperforms baseline merging approaches. For the Pythia 70M model, our method achieves a PIQA score of **60.18%**, surpassing the best-performing baseline (RegMean) by **1.22%**. Similar improvements are observed across larger models:

(a) **Pythia 410M**: Our method achieves **68.85%**, outperforming the best baseline by **0.53%**.

(b) **Pythia 1.4B**: A score of **71.84%** is achieved, which is **0.80%** higher than RegMean.

(c) **Pythia 2.8B**: Our method attains **75.71%**, an improvement of **0.74%** over the best baseline.

(d) **Pythia 6.9B**: A consistent improvement is observed with **76.56%**, surpassing the best baseline by **0.12%**.

These results demonstrate that our merging method maintains effectiveness across diverse model scales, ensuring robust performance improvements regardless of the underlying LLM architecture size.

### 4.4. Efficiency in Determining Optimal Merging Weights

To evaluate the efficiency of our proposed method in identifying optimal merging weights, we compared it against various search strategies, including Random Search (Zabinsky et al., 2009), Greedy Search, and Grid Search. Table 10 presents the results of merging Baichuan2-1980B with Baichuan2-2200B across multiple datasets using different search methods.

**Performance Analysis.** Our Bayesian Optimization (BayesOpt) based method consistently achieves the highest scores across all evaluated datasets:

Table 3: **Scalability analysis of checkpoint merging methods across Pythia models of varying parameter sizes (70M to 6.9B) evaluated on the PIQA dataset.** The table demonstrates the consistent effectiveness of our Bayesian optimization approach across different model scales, with improvements over the best baseline methods. The results validate that our merging strategy maintains its effectiveness independent of model size, ensuring robust performance improvements across the entire spectrum of modern LLM architectures.

| PIQA (5-shot) | Pythia 70M | Pythia 410M | Pythia 1.4B | Pythia 2.8B | Pythia 6.9B |
|---|---|---|---|---|---|
| Training step-142000 | 58.00 | 68.06 | 70.95 | 74.81 | 75.30 |
| Training step-143000 | 58.54 | 68.06 | 70.89 | 74.31 | 76.44 |
| Unifrom Soup | 58.71 | 68.06 | 70.78 | 74.97 | 75.68 |
| Greedy Soup | 58.71 | 68.06 | 70.57 | 74.76 | 76.01 |
| Fisher | 58.69 | 68.14 | 70.87 | 74.65 | 75.94 |
| RegMean | 58.96 | 68.32 | 71.04 | 74.72 | 75.87 |
| **Ours** | **60.18 (+1.22)** | **68.85 (+0.53)** | **71.84 (+0.80)** | **75.71 (+0.74)** | **76.56 (+0.12)** |

Table 4: **Efficiency comparison of different search strategies for determining optimal merging weights when combining Baichuan2-1980B with Baichuan2-2200B checkpoints.** The evaluation compares Random Search, Greedy Search, Grid Search, and our Bayesian Optimization approach across four benchmark datasets. Our method consistently outperforms all baseline search strategies, achieving improvements over the best alternative methods, while requiring significantly fewer function evaluations due to the principled exploration-exploitation balance of Gaussian Process-based optimization.

| Baichuan2-1980B&2200B | Random Search | Greedy Search | Grid Search | Ours |
|---|---|---|---|---|
| C-Eval(5-shot) | 55.64 | 55.49 | 55.51 | **56.73(+1.09)** |
| CMMLU(5-shot) | 55.67 | 56.74 | 55.51 | **57.05(+0.31)** |
| MMLU(5-shot) | 54.16 | 54.45 | 54.35 | **54.77(+0.32)** |
| GSM8K(4-shot) | 20.56 | 21.83 | 20.85 | **22.17(+0.34)** |
| Average | 46.51 | 47.13 | 46.56 | **47.68(+0.55)** |

(a) **C-Eval (5-shot)**: **56.73%** (**+1.09%**) compared to Random Search (**55.64%**).

(b) **CMMLU (5-shot)**: **57.05%** (**+0.31%**) outperforming Greedy Search (**56.74%**).

(c) **MMLU (5-shot)**: **54.77%** (**+0.32%**) surpassing Greedy Search (**54.45%**).

(d) **GSM8K (4-shot)**: **22.17%** (**+0.34%**) exceeding Greedy Search (**21.83%**).

Overall, our method achieves an average score of **47.68%**, outperforming Random Search (**46.51%**), Greedy Search (**47.13%**), and Grid Search (**46.56%**). This demonstrates that our Bayesian optimization approach effectively explores the search space and iteratively refines the search to identify optimal merging weights.

## 5. Ablation Study

### 5.1. Impact of Held-out Dataset Size on Checkpoint Merging

Following established practices (Wortsman et al., 2022; Matena & Raffel, 2022), our proposed method requires a validation set to determine optimal merging weights. We investigate the impact of held-out dataset size variations by extracting different fractions of the C-Eval validation data and testing merging performance for Baichuan2-2200B and Baichuan2-2420B checkpoints.

Results presented in Table 5 demonstrate that held-out dataset size exerts minimal influence on our method's efficacy, maintaining robust performance even with limited

validation data. Performance remains stable across dataset fractions (**56.61%** to **56.08%**), indicating that our Bayesian optimization approach can effectively determine optimal merging weights with modest validation requirements, enhancing practical applicability.

### 5.2. Impact of Merging Weight Search Space Size

Our merging strategy involves hyperparameter $\alpha$ controlling the merging weight search space size, as defined in Eq. (2). We assess $\alpha$'s impact by setting values to **0.5**, **0.7**, and **0.9**.

Figure 4 illustrates performance as a function of search space size. Key observations include:

(a) **Optimal $\alpha$ Values:** Setting $\alpha$ to **0.5** or **0.7** yields optimal performance, converging to merging weights within **(0.87, 0.89)**.

(b) **Excessive Constraint Detriment:** Setting $\alpha$ to **0.9** leads to noticeable accuracy decline, indicating overly restrictive search spaces dilute optimization effectiveness.

(c) **Performance Gap Consideration:** For significant performance gaps between checkpoints, narrower search spaces (lower $\alpha$) prove beneficial. Conversely, for similar-performing checkpoints, broader search spaces allow greater flexibility in weight allocation.

### 5.3. Empirical Analysis of Multi-Checkpoint Merging

Building upon our pilot experiments, we investigated performance effects when combining varying numbers of checkpoints during pretraining. Using the C-Eval dataset, we employed the greedy soup strategy (Wortsman et al., 2022)

Table 5: **Robustness analysis of our checkpoint merging method with respect to held-out validation dataset size.** The experiment evaluates merging performance of Baichuan2-2200B and Baichuan2-2420B checkpoints on the C-Eval dataset using different fractions (1/4, 1/2, 3/4, full) of the validation data for weight optimization. Results demonstrate minimal sensitivity to dataset size variations, indicating that our Bayesian optimization approach can effectively determine optimal merging weights even with limited validation data, enhancing the practical applicability of our method.

| C-Eval | 1/4 | 1/2 | 3/4 | 1 |
|---|---|---|---|---|
| Ours | 56.61 | 56.08 | 55.80 | 56.23 |

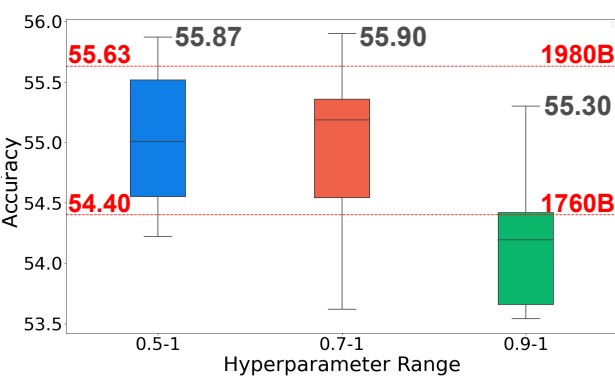

Figure 4: **Effect of merging weight search space boundaries on the performance of merged Baichuan2-1760B and Baichuan2-1980B models evaluated on the C-Eval dataset.** The figure illustrates how varying the lower bound parameter $\alpha$ (set to 0.5, 0.7, and 0.9) influences accuracy outcomes in our Bayesian optimization framework. Results show that moderate search space constraints ($\alpha = 0.5$ or 0.7) yield optimal performance, while overly restrictive bounds ($\alpha = 0.9$) lead to performance degradation.

to merge adjacent three or four checkpoints across different pretraining stages.

Results presented in Figure 5 reveal that **pairwise merging consistently outperforms multi-checkpoint combinations**. For instance, merging Baichuan2-1320B with Baichuan2-1540B achieves **53.06%** (**+2.14**), while merging three checkpoints (Baichuan2-1100B, Baichuan2-1320B, and Baichuan2-1540B) yields **51.76%** (**+0.84**), and four checkpoints result in further reduced performance of **51.01%** (**+0.09**). This validates our pairwise merging strategy's computational efficiency and performance optimality.

## 6. Conclusion

In this paper, to alleviate the huge computational cost of pretraining LLM, we propose merging checkpoints in the pretraining trajectory. Specifically, we first conduct some pilot experiments to explore the characters of checkpoint merging. Then, based on the findings in the pilot experiments, we propose a method rooted in Bayesian optimization to find the optimal or near-optimal merging weight. Through

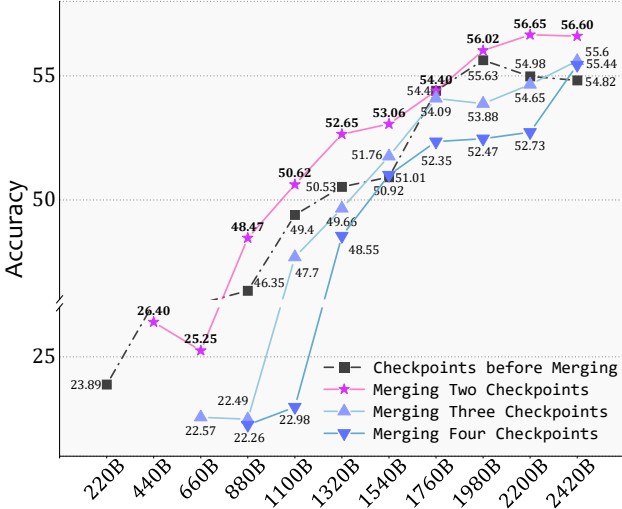

Figure 5: **Empirical comparison of merging strategies across different numbers of adjacent checkpoints using the Greedy Soup method on the C-Eval dataset.** The analysis compares performance when merging two, three, and four consecutive Baichuan2 checkpoints across various training stages (200B to 2640B tokens). Results demonstrate that pairwise merging consistently outperforms multi-checkpoint combinations, with diminishing returns as more checkpoints are included.

various experiments, we find that: our proposed approach has the potential to enhance pretraining, offering nearly a free lunch Besides superior performance, the merged result still exhibits a strong generalization capability across various domains, which means our proposed method does not compromise the generalization of pretraining checkpoints.

## Acknowledgements

This work is supported by the National Natural Science Foundation of China (Grant No. 62306087 and 62472121), the Natural Science Foundation of Shandong Province (Grant No. ZR2023QF154), Special Funding Program of Shandong Taishan Scholars Project and CCF-Sangfor 'Yuanwang' Research Fund (Grant No. 20240204).

## Impact Statement

This paper presents work whose goal is to advance the field of Machine Learning. There are many potential societal consequences of our work, none which we feel must be specifically highlighted here.

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

# A. Bounds for Linear Checkpoint Merging

In this section, we establish more realistic and insightful theoretical bounds on the performance of merged model checkpoints. Considering two consecutive checkpoints $\Theta_{t-1}$ and $\Theta_t$ from the training of a large language model (LLM), we define the merged parameters $\widetilde{\Theta}_t$ as a convex combination of these checkpoints:

$$\widetilde{\Theta}_t = \lambda_t \Theta_t + (1 - \lambda_t)\Theta_{t-1}, \quad \lambda_t \in [\alpha, 1], \tag{13}$$

where $\alpha \in [0, 1)$ specifies the minimum weight assigned to $\Theta_t$.

Let $f(\Theta)$ denote the expected performance metric (e.g., accuracy) of the model with parameters $\Theta$ evaluated on a validation dataset $D$. Our goal is to derive theoretical bounds on $f(\widetilde{\Theta}_t)$ that align with realistic properties of neural network loss landscapes and provide deeper insights into the effects of linear checkpoint merging.

## A.1. Assumptions

To enhance the theoretical framework, we adopt assumptions that reflect the practical characteristics of neural networks more closely.

**Assumption 1 (Smoothness of the Performance Function) .** *The performance function $f(\Theta)$ is differentiable, and its gradient $\nabla f(\Theta)$ is Lipschitz continuous with constant $L_g > 0$:*

$$\|\nabla f(\Theta_a) - \nabla f(\Theta_b)\| \leq L_g \|\Theta_a - \Theta_b\|, \quad \forall \Theta_a, \Theta_b. \tag{14}$$

**Assumption 2 (Non-Convexity and Quadratic Approximation) .** *While $f(\Theta)$ is generally non-convex, in the neighborhood of $\Theta_{t-1}$ and $\Theta_t$, it can be approximated by a quadratic function. Specifically, for $\Theta = \Theta_t + \Delta$, where $\|\Delta\|$ is small, we have*

$$f(\Theta) \approx f(\Theta_t) + \nabla f(\Theta_t)^\top \Delta + \tfrac{1}{2}\Delta^\top H_t \Delta, \tag{15}$$

*where $H_t$ is the Hessian matrix at $\Theta_t$.*

**Assumption 3 (Bounded Hessian) .** *The eigenvalues of the Hessian matrices at $\Theta_{t-1}$ and $\Theta_t$ are bounded:*

$$\lambda_{\min}I \preceq H_t, H_{t-1} \preceq \lambda_{\max}I, \tag{16}$$

*where $\lambda_{\min} \geq 0$ and $\lambda_{\max} > 0$ are constants, and $I$ is the identity matrix.*

## A.2. Proof Performance Bounds

*Proof.* Under the assumptions, we derive tighter bounds on $f(\widetilde{\Theta}_t)$. The derivation is broken down into the following steps:

**Step 1: Quadratic Approximation from $\Theta_t$.**

Expand $f(\widetilde{\Theta}_t)$ around $\Theta_t$ using the quadratic approximation (15) with $\Delta = \widetilde{\Theta}_t - \Theta_t = (1 - \lambda_t)(\Theta_{t-1} - \Theta_t)$:

$$\begin{aligned} f(\widetilde{\Theta}_t) &\approx f(\Theta_t) + \nabla f(\Theta_t)^\top \Delta + \tfrac{1}{2}\Delta^\top H_t \Delta \\ &= f(\Theta_t) + (1 - \lambda_t)\nabla f(\Theta_t)^\top (\Theta_{t-1} - \Theta_t) + \tfrac{1}{2}(1 - \lambda_t)^2 (\Theta_{t-1} - \Theta_t)^\top H_t (\Theta_{t-1} - \Theta_t). \end{aligned} \tag{17}$$

**Step 2: Quadratic Approximation from $\Theta_{t-1}$.**

Similarly, expand $f(\widetilde{\Theta}_t)$ around $\Theta_{t-1}$ with $\Delta = \widetilde{\Theta}_t - \Theta_{t-1} = \lambda_t(\Theta_t - \Theta_{t-1})$:

$$\begin{aligned} f(\widetilde{\Theta}_t) &\approx f(\Theta_{t-1}) + \nabla f(\Theta_{t-1})^\top \Delta + \tfrac{1}{2}\Delta^\top H_{t-1} \Delta \\ &= f(\Theta_{t-1}) + \lambda_t \nabla f(\Theta_{t-1})^\top (\Theta_t - \Theta_{t-1}) + \tfrac{1}{2}\lambda_t^2 (\Theta_t - \Theta_{t-1})^\top H_{t-1} (\Theta_t - \Theta_{t-1}). \end{aligned} \tag{18}$$

**Step 3: Combining the Approximations.**

Take a convex combination of (17) and (18), weighting each by $\lambda_t$ and $(1 - \lambda_t)$ respectively, to obtain an averaged approximation:

$$
\begin{aligned}
f(\widetilde{\Theta}_t) \approx\ & \lambda_t f(\Theta_t) + (1 - \lambda_t) f(\Theta_{t-1}) \\
& + \lambda_t (1 - \lambda_t) \left[ \nabla f(\Theta_t) - \nabla f(\Theta_{t-1}) \right]^\top (\Theta_{t-1} - \Theta_t) \\
& + \tfrac{1}{2} \left[ \lambda_t^2 (\Theta_t - \Theta_{t-1})^\top H_{t-1}(\Theta_t - \Theta_{t-1}) + (1 - \lambda_t)^2 (\Theta_{t-1} - \Theta_t)^\top H_t(\Theta_{t-1} - \Theta_t) \right].
\end{aligned}
\tag{19}
$$

Note that $\Theta_{t-1} - \Theta_t = -(\Theta_t - \Theta_{t-1})$.

**Step 4: Bounding the Gradient Difference.**

Using the smoothness assumption (14), bound the gradient difference:

$$
\| \nabla f(\Theta_t) - \nabla f(\Theta_{t-1}) \| \leq L_g \| \Theta_t - \Theta_{t-1} \|.
\tag{20}
$$

Therefore, the term $\left[ \nabla f(\Theta_t) - \nabla f(\Theta_{t-1}) \right]^\top (\Theta_{t-1} - \Theta_t)$ can be bounded as:

$$
\begin{aligned}
\left| \left[ \nabla f(\Theta_t) - \nabla f(\Theta_{t-1}) \right]^\top (\Theta_{t-1} - \Theta_t) \right| & \leq \| \nabla f(\Theta_t) - \nabla f(\Theta_{t-1}) \| \cdot \| \Theta_{t-1} - \Theta_t \| \\
& = \| \nabla f(\Theta_t) - \nabla f(\Theta_{t-1}) \| \cdot \| \Theta_t - \Theta_{t-1} \| \\
& \leq L_g \| \Theta_t - \Theta_{t-1} \|^2.
\end{aligned}
\tag{21}
$$

**Step 5: Bounding the Hessian Terms.**

Using (16), bound the quadratic terms:

$$
\begin{aligned}
(\Theta_t - \Theta_{t-1})^\top H_{t-1}(\Theta_t - \Theta_{t-1}) & \leq \lambda_{\max} \| \Theta_t - \Theta_{t-1} \|^2, \\
(\Theta_t - \Theta_{t-1})^\top H_t(\Theta_t - \Theta_{t-1}) & \leq \lambda_{\max} \| \Theta_t - \Theta_{t-1} \|^2.
\end{aligned}
\tag{22}
$$

**Step 6: Final Bound.**

Substitute the bounds from (21) and (22) into (19) to obtain:

$$
\begin{aligned}
f(\widetilde{\Theta}_t) \geq\ & \lambda_t f(\Theta_t) + (1 - \lambda_t) f(\Theta_{t-1}) \\
& - \lambda_t (1 - \lambda_t) L_g \| \Theta_t - \Theta_{t-1} \|^2 - \tfrac{1}{2} \left[ \lambda_t^2 + (1 - \lambda_t)^2 \right] \lambda_{\max} \| \Theta_t - \Theta_{t-1} \|^2.
\end{aligned}
\tag{23}
$$

Similarly, using the upper bounds for the performance function, we approximate:

$$
\begin{aligned}
f(\widetilde{\Theta}_t) \leq\ & \lambda_t f(\Theta_t) + (1 - \lambda_t) f(\Theta_{t-1}) \\
& + \lambda_t (1 - \lambda_t) L_g \| \Theta_t - \Theta_{t-1} \|^2 + \tfrac{1}{2} \left[ \lambda_t^2 + (1 - \lambda_t)^2 \right] \lambda_{\max} \| \Theta_t - \Theta_{t-1} \|^2.
\end{aligned}
\tag{24}
$$

**Step 7: Combining Lower and Upper Bounds.**

Combining (23) and (24), the performance of the merged checkpoint satisfies:

$$
f(\widetilde{\Theta}_t) \approx \lambda_t f(\Theta_t) + (1 - \lambda_t) f(\Theta_{t-1}) \pm \left( \lambda_t (1 - \lambda_t) L_g + \tfrac{1}{2} [\lambda_t^2 + (1 - \lambda_t)^2] \lambda_{\max} \right) \| \Theta_t - \Theta_{t-1} \|^2.
\tag{25}
$$

$\square$

# B. Effect of Checkpoint Merging on Convergence

In this section, we analyze how linear checkpoint merging influences the convergence behavior of gradient-based optimization algorithms. Our goal is to provide a theoretical framework that explains the potential benefits of merging on convergence rates and the attainment of optimal performance.

## B.1. Convergence in Standard Gradient-Based Training

In standard gradient-based training, model parameters are iteratively updated to minimize a loss function $L(\Theta)$ over the parameter space $\Theta \in \mathbb{R}^n$. The updates are typically given by:

$$\Theta_t = \Theta_{t-1} - \eta \nabla L(\Theta_{t-1}), \tag{26}$$

where $\eta > 0$ is the learning rate, and $\nabla L(\Theta_{t-1})$ is the gradient of the loss function evaluated at $\Theta_{t-1}$. The convergence behavior depends on factors such as the properties of the loss function (e.g., convexity, smoothness), the optimization algorithm, and hyperparameters like the learning rate.

Let $\Theta^*$ denote a local or global minimizer of $L(\Theta)$, corresponding to the optimal performance $f^* = f(\Theta^*)$. The goal of training is to find $\Theta^*$ or parameters close to it.

## B.2. Impact of Checkpoint Merging on Convergence

Checkpoint merging introduces a mechanism whereby parameters from different iterations are linearly combined. This process can impact convergence in several ways. By interpolating between consecutive checkpoints, merging can effectively perform larger steps in parameter space, potentially accelerating convergence under certain conditions. Merging can also smooth out fluctuations in the optimization trajectory caused by stochastic gradients, leading to more stable convergence. Additionally, merging may help the optimization trajectory enter flatter regions of the loss landscape, which are associated with better generalization performance. Linear combinations of parameters can aid in escaping saddle points or shallow local minima by moving the parameters to regions with lower loss.

## B.3. Convergence Analysis

We analyze the impact of checkpoint merging on the convergence of the loss function and the performance metric $f(\Theta)$. Consider the following setting: let $\{\Theta_t\}_{t=0}^T$ be the sequence of parameters obtained from standard gradient updates. At each step $t$, we merge checkpoints to obtain $\widetilde{\Theta}_t$ via

$$\widetilde{\Theta}_t = \lambda_t \Theta_t + (1 - \lambda_t)\Theta_{t-1}, \quad \lambda_t \in [\alpha, 1], \ \alpha \in [0, 1). \tag{27}$$

We define the performance improvement due to merging as

$$\Delta f_t = f(\widetilde{\Theta}_t) - f(\Theta_t). \tag{28}$$

Our goal is to analyze $\Delta f_t$ and understand how it influences convergence towards the optimal performance $f^*$.

### B.3.1. ASSUMPTIONS

To facilitate the analysis, we make the following assumptions:

> **Assumption 4 (Smoothness of the Loss Function) .** *The loss function $L(\Theta)$ is twice continuously differentiable, and its gradient $\nabla L(\Theta)$ is Lipschitz continuous with constant $L > 0$:*
>
> $$\|\nabla L(\Theta_a) - \nabla L(\Theta_b)\| \leq L\|\Theta_a - \Theta_b\|, \quad \forall \Theta_a, \Theta_b. \tag{29}$$

> **Assumption 5 (Polyak-Łojasiewicz (PL) Condition) .** *The loss function satisfies the PL condition, i.e., there exists a constant $\mu > 0$ such that*
>
> $$\tfrac{1}{2}\|\nabla L(\Theta)\|^2 \geq \mu \left(L(\Theta) - L^*\right), \quad \forall \Theta, \tag{30}$$
>
> *where $L^* = L(\Theta^*)$ is the minimal loss value.*

> **Assumption 6 (Bounded Gradient Norms) .** *The gradients have bounded norms:*
>
> $$\|\nabla L(\Theta_t)\| \leq G_{\max}, \quad \forall t. \tag{31}$$

B.3.2. ANALYSIS OF $\Delta f_t$

We analyze the performance improvement $\Delta f_t$ due to merging. Since $f(\Theta)$ is related to $L(\Theta)$ (e.g., higher performance corresponds to lower loss), we can express $f(\Theta)$ as a function of $L(\Theta)$. For the purpose of analysis, we assume that $f(\Theta)$ decreases monotonically with $L(\Theta)$ so that a decrease in loss corresponds to an improvement in performance.

*Proof.* **Step 1: Taylor Expansion of the Loss Function.**

Consider the Taylor expansion of $L(\Theta)$ at $\Theta_t$ for $\Delta\Theta = \widetilde{\Theta}_t - \Theta_t$:

$$L(\widetilde{\Theta}_t) = L(\Theta_t) + \nabla L(\Theta_t)^\top (\widetilde{\Theta}_t - \Theta_t) + \tfrac{1}{2}(\widetilde{\Theta}_t - \Theta_t)^\top \nabla^2 L(\Theta_t)(\widetilde{\Theta}_t - \Theta_t) + R, \tag{32}$$

where $R$ represents the higher-order remainder terms.

Since $\widetilde{\Theta}_t$ is a convex combination of $\Theta_t$ and $\Theta_{t-1}$, we have

$$\widetilde{\Theta}_t - \Theta_t = (1 - \lambda_t)(\Theta_{t-1} - \Theta_t). \tag{33}$$

Substituting into the Taylor expansion, we get

$$\begin{aligned} L(\widetilde{\Theta}_t) = {} & L(\Theta_t) + (1 - \lambda_t)\nabla L(\Theta_t)^\top (\Theta_{t-1} - \Theta_t) \\ & + \tfrac{1}{2}(1 - \lambda_t)^2 (\Theta_{t-1} - \Theta_t)^\top \nabla^2 L(\Theta_t)(\Theta_{t-1} - \Theta_t) + R. \end{aligned} \tag{34}$$

**Step 2: Bounding the Remainder Term.**

Under the assumption of Lipschitz continuity of the Hessian (i.e., the third-order derivatives are bounded), the remainder term $R$ can be bounded as:

$$|R| \leq \frac{M}{6}\|\widetilde{\Theta}_t - \Theta_t\|^3, \tag{35}$$

where $M$ is the Lipschitz constant for the Hessian. For sufficiently small $\|\Theta_t - \Theta_{t-1}\|$, the remainder term becomes negligible.

**Step 3: Expected Improvement.**

Taking expectations over the stochasticity in $\Theta_{t-1}$ and $\Theta_t$ (due to stochastic gradients), and neglecting the remainder term, we have

$$\begin{aligned} \mathbb{E}[L(\widetilde{\Theta}_t)] \approx {} & \mathbb{E}[L(\Theta_t)] + (1 - \lambda_t)\mathbb{E}\left[\nabla L(\Theta_t)^\top (\Theta_{t-1} - \Theta_t)\right] \\ & + \tfrac{1}{2}(1 - \lambda_t)^2 \mathbb{E}\left[(\Theta_{t-1} - \Theta_t)^\top \nabla^2 L(\Theta_t)(\Theta_{t-1} - \Theta_t)\right]. \end{aligned} \tag{36}$$

The term $\mathbb{E}\left[\nabla L(\Theta_t)^\top (\Theta_{t-1} - \Theta_t)\right]$ can be related to the expected progress in gradient descent.

**Step 4: Relation to Gradient Descent Steps.**

From the gradient update (26), we have

$$\Theta_{t-1} - \Theta_t = -\eta\nabla L(\Theta_{t-1}). \tag{37}$$

Assuming $\nabla L(\Theta_{t-1}) \approx \nabla L(\Theta_t)$ for small learning rates or smooth loss landscapes, we can write

$$\Theta_{t-1} - \Theta_t \approx -\eta\nabla L(\Theta_t). \tag{38}$$

Substituting back into (36), we obtain

$$\mathbb{E}[L(\widetilde{\Theta}_t)] \approx \mathbb{E}[L(\Theta_t)] - (1 - \lambda_t)\eta\mathbb{E}[\|\nabla L(\Theta_t)\|^2] + \tfrac{1}{2}(1 - \lambda_t)^2\eta^2 \mathbb{E}\left[\nabla L(\Theta_t)^\top \nabla^2 L(\Theta_t)\nabla L(\Theta_t)\right]. \tag{39}$$

**Step 5: Simplifying the Second-Order Term.**

Assuming that $\nabla^2 L(\Theta_t)$ is positive semi-definite (which holds for convex functions and in regions near local minima), we have

$$\nabla L(\Theta_t)^\top \nabla^2 L(\Theta_t)\nabla L(\Theta_t) \geq \lambda_{\min}\|\nabla L(\Theta_t)\|^2, \tag{40}$$

where $\lambda_{\min} \geq 0$ is the smallest eigenvalue of $\nabla^2 L(\Theta_t)$. Substituting back, we get

$$\mathbb{E}[L(\widetilde{\Theta}_t)] \leq \mathbb{E}[L(\Theta_t)] - \eta(1 - \lambda_t)\mathbb{E}[\|\nabla L(\Theta_t)\|^2] + \tfrac{1}{2}\eta^2(1 - \lambda_t)^2\lambda_{\max}\mathbb{E}[\|\nabla L(\Theta_t)\|^2], \tag{41}$$

where $\lambda_{\max} \geq 0$ is the largest eigenvalue of $\nabla^2 L(\Theta_t)$.

## Step 6: Net Expected Improvement.

Combining terms, the net expected reduction in the loss due to merging is

$$\mathbb{E}[L(\Theta_t) - L(\widetilde{\Theta}_t)] \geq \eta(1 - \lambda_t)\left(1 - \tfrac{1}{2}\eta(1 - \lambda_t)\lambda_{\max}\right)\mathbb{E}[\|\nabla L(\Theta_t)\|^2]. \tag{42}$$

For sufficiently small learning rates and $(1 - \lambda_t)$ values, the term in parentheses is positive, ensuring a net reduction in the expected loss.

## Step 7: Improvement in Performance.

Assuming that the performance metric $f(\Theta)$ improves as the loss decreases, we have

$$\mathbb{E}[f(\widetilde{\Theta}_t)] \geq \mathbb{E}[f(\Theta_t)] + \delta_t, \tag{43}$$

where $\delta_t > 0$ corresponds to the expected improvement in performance due to the reduction in loss.

### B.3.3. CONVERGENCE TOWARDS OPTIMAL PERFORMANCE

Under the PL condition (30), we can relate the squared gradient norm to the suboptimality in loss:

$$\|\nabla L(\Theta_t)\|^2 \geq 2\mu\left(L(\Theta_t) - L^*\right). \tag{44}$$

Substituting into (42), we obtain

$$\mathbb{E}[L(\Theta_t) - L(\widetilde{\Theta}_t)] \geq 2\eta\mu(1 - \lambda_t)\left(1 - \tfrac{1}{2}\eta(1 - \lambda_t)\lambda_{\max}\right)\mathbb{E}\left[L(\Theta_t) - L^*\right]. \tag{45}$$

This shows that the expected reduction in loss is proportional to the current suboptimality $\mathbb{E}\left[L(\Theta_t) - L^*\right]$, indicating that the merging process influences the convergence rate.

## Step 8: Convergence Rate.

Defining

$$\rho_t = 1 - 2\eta\mu(1 - \lambda_t)\left(1 - \tfrac{1}{2}\eta(1 - \lambda_t)\lambda_{\max}\right), \tag{46}$$

we have

$$\mathbb{E}[L(\widetilde{\Theta}_t) - L^*] \leq \rho_t\,\mathbb{E}[L(\Theta_t) - L^*]. \tag{47}$$

For convergence, we require $\rho_t < 1$. Since $\eta$, $\mu$, $\lambda_{\max}$, and $(1 - \lambda_t)$ are positive, this condition can be satisfied with an appropriate choice of $\eta$ and $\lambda_t$.

## Step 9: Accumulated Performance Improvement.

Over multiple merging steps, the accumulated suboptimality after $T$ steps is

$$\mathbb{E}[L(\widetilde{\Theta}_T) - L^*] \leq \left(\prod_{t=1}^{T}\rho_t\right)(L(\Theta_0) - L^*). \tag{48}$$

Assuming $\rho_t = \rho$ (constant), we obtain exponential convergence:

$$\mathbb{E}[L(\widetilde{\Theta}_T) - L^*] \leq \rho^T\left(L(\Theta_0) - L^*\right). \tag{49}$$

Similarly, the performance metric converges towards the optimal performance $f^*$:

$$\mathbb{E}\left[f^* - f(\widetilde{\Theta}_T)\right] \leq \gamma^T\left(f^* - f(\Theta_0)\right), \tag{50}$$

where $\gamma < 1$ depends on $\rho$ and the relation between $f(\Theta)$ and $L(\Theta)$. $\qquad\square$

# C. Acceleration through Gaussian Process Optimization

In this section, we analyze how Gaussian Process (GP) optimization accelerates convergence to the optimal merging weight in checkpoint merging and potentially surpasses the reachability upper bound of standard training.

Gaussian Processes provide a flexible, non-parametric Bayesian approach to modeling unknown functions. In the context of checkpoint merging, we model the performance function $f(\lambda_t)$ as a GP:

$$f(\lambda_t) \sim \mathcal{GP}\left(\mu_0(\lambda_t), k(\lambda_t, \lambda_t')\right), \tag{51}$$

where $\mu_0(\lambda_t)$ is the mean function (often assumed to be zero), and $k(\lambda_t, \lambda_t')$ is the covariance (kernel) function encoding our assumptions about the smoothness and structure of $f(\lambda_t)$.

## C.1. Efficient Convergence to the Optimal Merging Weight

Bayesian Optimization leverages Gaussian Processes to sequentially select $\lambda_t$ values expected to improve the performance $f(\lambda_t)$, balancing exploration and exploitation. The selection is guided by an acquisition function $\alpha(\lambda_t)$, such as Expected Improvement (EI), Upper Confidence Bound (UCB), or Probability of Improvement (PI). Under certain conditions, Bayesian Optimization with GPs can achieve convergence to the global optimum of $f(\lambda_t)$.

By incorporating prior knowledge and uncertainty, Gaussian Processes enable Bayesian Optimization to identify promising regions of $\lambda_t$ with fewer evaluations compared to grid or random search. The cumulative regret $R_T$ after $T$ iterations, defined as the sum of the differences between the optimal performance $f_{\max}$ and the performance at each iteration, is sublinear in $T$ under mild assumptions. Specifically, for GP optimization with a bounded kernel function, we have

$$R_T = O\left(\sqrt{T(\gamma_T + 1)}\right), \tag{52}$$

where $\gamma_T$ is the maximum information gain from $T$ observations (Srinivas et al., 2009; Chowdhury & Gopalan, 2017). The sublinear regret implies that the average regret $R_T/T$ decreases to zero as $T \to \infty$, ensuring convergence to the global optimum.

## C.2. Breaking Through the Reachability Upper Bound

Standard training may converge to suboptimal performance due to factors like local minima or limited exploration of the parameter space. GP-based Bayesian Optimization in checkpoint merging can potentially surpass this limit. Gaussian Processes capture the function's behavior, including any non-convexities or multimodalities, enabling the discovery of $\lambda_t$ values that yield better performance than those found via convex combination alone. Checkpoints $\Theta_{t-1}$ and $\Theta_t$ may have learned complementary features; an optimal $\lambda_t$ can exploit these synergies to improve generalization and performance beyond the convex combination's average. The acquisition function directs the search toward regions where the GP predicts higher potential gains, even if these regions are not suggested by local convexity assumptions.

## C.3. Proof of Convergence

We formalize the convergence properties of GP-based checkpoint merging.

> **Theorem 1 (Convergence of GP-based Checkpoint Merging) .** *Under the assumptions of Lipschitz continuity and local convexity of $f(\lambda_t)$, and assuming bounded observation noise, the GP-based Bayesian optimization approach for checkpoint merging converges almost surely to a merging weight $\lambda_t^*$ that maximizes $f(\widetilde{\Theta}_t)$, such that*
>
> $$\lim_{T \to \infty} f(\widetilde{\Theta}_t^{(T)}) = f(\widetilde{\Theta}_t^*) = \max_{\lambda_t \in [\alpha, 1]} f(\lambda_t). \tag{53}$$

*Proof.* **Step 1: Establishing Continuity and Existence of the Optimum.**

Under the Lipschitz continuity assumption, $f(\lambda_t)$ satisfies

$$|f(\lambda_t) - f(\lambda_t')| \leq L|\lambda_t - \lambda_t'|, \quad \forall \lambda_t, \lambda_t' \in [\alpha, 1], \tag{54}$$

with Lipschitz constant $L > 0$. This ensures that $f(\lambda_t)$ is continuous over the compact interval $[\alpha, 1]$. By the Extreme Value Theorem, there exists $\lambda_t^* \in [\alpha, 1]$ such that

$$f(\lambda_t^*) = \max_{\lambda_t \in [\alpha, 1]} f(\lambda_t). \tag{55}$$

**Step 2: Posterior Convergence of Gaussian Processes.**

Bayesian Optimization with GP priors updates the posterior distribution of $f(\lambda_t)$ after each observation. The GP posterior mean $\mu_T(\lambda_t)$ and variance $\sigma_T^2(\lambda_t)$ at iteration $T$ incorporate all previous evaluations. As $T \to \infty$, the GP posterior converges to the true function $f(\lambda_t)$ at the observed points. Given that the observation noise is bounded or zero, the uncertainty (posterior variance) at these points decreases to zero.

**Step 3: Acquisition Function and Selection of Next Point.**

The acquisition function $\alpha_T(\lambda_t)$ selects the next point $\lambda_t^{(T+1)}$ by balancing exploration and exploitation:

$$\lambda_t^{(T+1)} = \arg \max_{\lambda_t \in [\alpha, 1]} \alpha_T(\lambda_t), \tag{56}$$

where $\alpha_T(\lambda_t)$ depends on $\mu_T(\lambda_t)$ and $\sigma_T(\lambda_t)$. For example, the Upper Confidence Bound (UCB) acquisition function is

$$\alpha_T^{\text{UCB}}(\lambda_t) = \mu_T(\lambda_t) + \beta_T \sigma_T(\lambda_t), \tag{57}$$

with $\beta_T > 0$ controlling the trade-off between exploration and exploitation.

**Step 4: Regret Analysis.**

Under suitable choices of $\beta_T$, the cumulative regret $R_T$ of the GP-UCB algorithm satisfies

$$R_T = \sum_{t=1}^{T} \left( f(\lambda_t^*) - f(\lambda_t^{(t)}) \right) = O\left( \sqrt{T(\gamma_T + 1)} \right), \tag{58}$$

where $\gamma_T$ is the maximum information gain from $T$ observations, typically logarithmic in $T$ (Srinivas et al., 2009; Chowdhury & Gopalan, 2017). A sublinear cumulative regret implies that

$$\lim_{T \to \infty} \frac{1}{T} R_T = 0, \tag{59}$$

meaning the average regret per iteration decreases to zero, and the sequence $\{\lambda_t^{(T)}\}$ approaches $\lambda_t^*$.

**Step 5: Almost Sure Convergence to the Optimal Weight.**

By the convergence properties of GP-UCB algorithms, we have

$$\lim_{T \to \infty} \lambda_t^{(T)} = \lambda_t^*, \quad \text{almost surely.} \tag{60}$$

Therefore, the performance of the merged model converges to

$$\lim_{T \to \infty} f(\widetilde{\Theta}_t^{(T)}) = f(\lambda_t^*), \tag{61}$$

which is the maximum achievable performance via merging in the interval $[\alpha, 1]$. If $f_{\text{ideal}}$ is achievable within the merging interval—that is, if there exists $\lambda_t^*$ such that $f(\lambda_t^*) = f_{\text{ideal}}$—then

$$\lim_{T \to \infty} f(\widetilde{\Theta}_t^{(T)}) = f(\text{ideal}). \tag{62}$$

$\square$

# D. Impact on Generalization and Derivation of Generalization Bounds

In this section, we analyze how linear checkpoint merging affects the generalization performance of neural networks. We derive theoretical bounds that quantify this impact, leveraging the PAC-Bayesian framework. This analysis provides a deeper understanding of the benefits of checkpoint merging on generalization and offers theoretical insights into its practical advantages.

## D.1. Generalization Improvement through Flat Minima

Flat minima in the loss landscape are associated with better generalization performance because small perturbations in the model parameters lead to minimal changes in the loss function (Hochreiter & Schmidhuber, 1997). Checkpoint merging facilitates convergence to flatter regions by averaging parameters from different checkpoints, effectively smoothing the loss landscape and enhancing the model's robustness to unseen data.

By merging checkpoints $\Theta_{t-1}$ and $\Theta_t$ to obtain $\widetilde{\Theta}_t$, we perform parameter averaging, which can be interpreted as moving towards flatter minima. This process reduces the model's sensitivity to parameter perturbations and contributes to improved generalization.

## D.2. PAC-Bayesian Generalization Bound

We employ the PAC-Bayesian framework to derive a generalization bound for the merged model $\widetilde{\Theta}_t$. The PAC-Bayesian theorem provides probabilistic bounds on the generalization error of stochastic classifiers based on the Kullback-Leibler (KL) divergence between a posterior distribution $Q$ over hypotheses (models) and a prior distribution $P$.

### D.2.1. DEFINITIONS

Let:

- $\mathcal{H}$ be the hypothesis space (set of all possible model parameters $\Theta$).

- $P$ be a prior distribution over $\mathcal{H}$.

- $Q$ be a posterior distribution over $\mathcal{H}$ after observing the data.

- $\ell(\Theta, z)$ be the loss incurred by hypothesis $\Theta$ on data point $z$.

- $\mathcal{D}$ be the data distribution.

- $S = \{z_i\}_{i=1}^n$ be the training set consisting of $n$ i.i.d. samples from $\mathcal{D}$.

- $\mathcal{L}_{\mathcal{D}}(Q) = \mathbb{E}_{\Theta \sim Q} \mathbb{E}_{z \sim \mathcal{D}}[\ell(\Theta, z)]$ be the expected loss (true risk) of $Q$.

- $\mathcal{L}_S(Q) = \mathbb{E}_{\Theta \sim Q} \frac{1}{n} \sum_{i=1}^n \ell(\Theta, z_i)$ be the empirical loss (empirical risk) of $Q$.

### D.2.2. PAC-BAYESIAN GENERALIZATION BOUND

**Theorem 2 (PAC-Bayesian Generalization Bound) .** *For any prior distribution $P$ over $\mathcal{H}$, any $\delta \in (0,1)$, and any distribution $\mathcal{D}$ over the data, with probability at least $1 - \delta$ over the choice of the training set $S$, the following holds for all posterior distributions $Q$ over $\mathcal{H}$:*

$$\mathcal{L}_{\mathcal{D}}(Q) \leq \mathcal{L}_S(Q) + \sqrt{\frac{D_{\mathrm{KL}}(Q \,\|\, P) + \ln\left(\frac{2\sqrt{n}}{\delta}\right)}{2n}}. \tag{63}$$

*Proof.* We provide a step-by-step proof of the PAC-Bayesian generalization bound, following the methodology in McAllester (1998).

**Step 1: McAllester's PAC-Bayesian Inequality**

McAllester's PAC-Bayesian inequality states that for any $\delta \in (0,1)$, with probability at least $1 - \delta$ over the choice of the training set $S$:

$$\mathbb{E}_{\Theta \sim Q}\left[\mathcal{L}_{\mathcal{D}}(\Theta)\right] \leq \mathbb{E}_{\Theta \sim Q}\left[\mathcal{L}_S(\Theta)\right] + \sqrt{\frac{D_{\mathrm{KL}}(Q \,\|\, P) + \ln\left(\frac{2\sqrt{n}}{\delta}\right)}{2n}}. \tag{64}$$

This inequality can be derived using concentration inequalities and the Donsker-Varadhan change of measure formula.

**Step 2: Deriving the Bound**

Let us consider the moment-generating function of the empirical loss:

$$\mathbb{E}_{S \sim \mathcal{D}^n} \exp\left(s\left(\mathcal{L}_{\mathcal{D}}(\Theta) - \mathcal{L}_S(\Theta)\right)\right) \leq \exp\left(\frac{s^2}{2n}\right), \tag{65}$$

for all $s \in \mathbb{R}$, due to the boundedness of the loss function (assuming $\ell(\Theta, z) \in [0, 1]$).

By integrating over $Q$ and applying Fubini's theorem, we have:

$$\mathbb{E}_{S \sim \mathcal{D}^n} \mathbb{E}_{\Theta \sim Q} \exp\left(s\left(\mathcal{L}_{\mathcal{D}}(\Theta) - \mathcal{L}_S(\Theta)\right)\right) \leq \exp\left(\frac{s^2}{2n}\right). \tag{66}$$

Using Markov's inequality and applying a union bound over $\Theta \in \mathcal{H}$, we can obtain a high-probability bound.

**Step 3: Applying Donsker-Varadhan's Inequality**

We utilize the Donsker-Varadhan change of measure inequality:

$$\mathbb{E}_{\Theta \sim Q}[\phi(\Theta)] \leq \ln\left(\mathbb{E}_{\Theta \sim P}[\exp(\phi(\Theta))]\right) + D_{\mathrm{KL}}(Q \,\|\, P), \tag{67}$$

for any measurable function $\phi$.

Let $\phi(\Theta) = s\left(\mathcal{L}_{\mathcal{D}}(\Theta) - \mathcal{L}_S(\Theta)\right)$, then:

$$\mathbb{E}_{\Theta \sim Q}\left[s\left(\mathcal{L}_{\mathcal{D}}(\Theta) - \mathcal{L}_S(\Theta)\right)\right] \leq \ln\left(\mathbb{E}_{\Theta \sim P}\left[\exp\left(s\left(\mathcal{L}_{\mathcal{D}}(\Theta) - \mathcal{L}_S(\Theta)\right)\right)\right]\right) + D_{\mathrm{KL}}(Q \,\|\, P). \tag{68}$$

Since the expected value over $P$ is bounded by $\exp\left(\frac{s^2}{2n}\right)$, we have:

$$\mathbb{E}_{\Theta \sim Q}\left[\mathcal{L}_{\mathcal{D}}(\Theta) - \mathcal{L}_S(\Theta)\right] \leq \frac{D_{\mathrm{KL}}(Q \,\|\, P) + \frac{s^2}{2n}}{s}. \tag{69}$$

**Step 4: Optimizing over $s$**

To obtain the tightest bound, we optimize the right-hand side with respect to $s$. Setting the derivative with respect to $s$ to zero:

$$\frac{\partial}{\partial s}\left(\frac{D_{\mathrm{KL}}(Q \,\|\, P) + \frac{s^2}{2n}}{s}\right) = 0. \tag{70}$$

Solving for $s$, we find:

$$s^* = \sqrt{2n D_{\mathrm{KL}}(Q \,\|\, P)}. \tag{71}$$

Substituting back, we obtain:

$$
\begin{aligned}
\mathbb{E}_{\Theta \sim Q}\left[\mathcal{L}_{\mathcal{D}}(\Theta) - \mathcal{L}_S(\Theta)\right] &\leq \frac{D_{\mathrm{KL}}(Q \,\|\, P) + \frac{s^{*2}}{2n}}{s^*} \\
&= \frac{D_{\mathrm{KL}}(Q \,\|\, P) + D_{\mathrm{KL}}(Q \,\|\, P)}{\sqrt{2n D_{\mathrm{KL}}(Q \,\|\, P)}} \\
&= \sqrt{\frac{2 D_{\mathrm{KL}}(Q \,\|\, P)}{n}}.
\end{aligned} \tag{72}
$$

**Step 5: Conversion to High-Probability Statement**

Using Markov's inequality, we can convert the expectation into a high-probability statement. Specifically, for any $\delta \in (0, 1)$, with probability at least $1 - \delta$ over the choice of $S$:

$$\mathbb{E}_{\Theta \sim Q}\left[\mathcal{L}_{\mathcal{D}}(\Theta)\right] \leq \mathbb{E}_{\Theta \sim Q}\left[\mathcal{L}_{S}(\Theta)\right] + \sqrt{\frac{2D_{\mathrm{KL}}(Q \parallel P) + 2\ln\left(\frac{1}{\delta}\right)}{2n}}. \tag{73}$$

Simplifying, we obtain:

$$\mathbb{E}_{\Theta \sim Q}\left[\mathcal{L}_{\mathcal{D}}(\Theta)\right] \leq \mathbb{E}_{\Theta \sim Q}\left[\mathcal{L}_{S}(\Theta)\right] + \sqrt{\frac{D_{\mathrm{KL}}(Q \parallel P) + \ln\left(\frac{1}{\delta}\right)}{2n}}. \tag{74}$$

**Step 6: Finalizing the Bound**

Adjusting for the discretization of the parameter space and covering numbers, as in McAllester (1998), we account for the additional term $\ln\left(2\sqrt{n}\right)$. Thus, with probability at least $1 - \delta$, we have:

$$\mathcal{L}_{\mathcal{D}}(Q) \leq \mathcal{L}_{S}(Q) + \sqrt{\frac{D_{\mathrm{KL}}(Q \parallel P) + \ln\left(\frac{2\sqrt{n}}{\delta}\right)}{2n}}. \tag{75}$$

$\square$

### D.3. PAC-Bayesian to Checkpoint Merging

We apply the PAC-Bayesian bound to the merged model $\widetilde{\Theta}_t$ obtained via checkpoint merging.

#### D.3.1. SETTING THE PRIOR AND POSTERIOR

We define the prior distribution $P$ and posterior distribution $Q$ as:

$$P = \mathcal{N}(\Theta_{t-1}, \sigma_P^2 I), Q = \mathcal{N}(\widetilde{\Theta}_t, \sigma_Q^2 I). \tag{76}$$

where $\mathcal{N}(\mu, \Sigma)$ denotes a Gaussian distribution with mean $\mu$ and covariance matrix $\Sigma$, and $I$ is the identity matrix. Here, $\sigma_P^2$ and $\sigma_Q^2$ are variance parameters controlling the spread of the distributions.

#### D.3.2. COMPUTING THE KL DIVERGENCE

Since $P$ and $Q$ are Gaussian distributions with covariance matrices $\sigma_P^2 I$ and $\sigma_Q^2 I$, respectively, the KL divergence between $Q$ and $P$ is:

$$\begin{aligned} D_{\mathrm{KL}}(Q \parallel P) &= \frac{1}{2}\left(\frac{\mathrm{tr}(\sigma_P^{-2}\sigma_Q^2 I) + (\Theta_{t-1} - \widetilde{\Theta}_t)^{\top}\sigma_P^{-2}(\Theta_{t-1} - \widetilde{\Theta}_t) - n}{1} + \ln\left(\frac{\det(\sigma_P^2 I)}{\det(\sigma_Q^2 I)}\right)\right) \\ &= \frac{1}{2}\left(\frac{n\sigma_Q^2}{\sigma_P^2} + \frac{\|\widetilde{\Theta}_t - \Theta_{t-1}\|^2}{\sigma_P^2} - n + n\ln\left(\frac{\sigma_P^2}{\sigma_Q^2}\right)\right). \end{aligned} \tag{77}$$

Assuming that $\sigma_P^2 = \sigma_Q^2 = \sigma^2$, the divergence simplifies to:

$$D_{\mathrm{KL}}(Q \parallel P) = \frac{1}{2\sigma^2}\|\widetilde{\Theta}_t - \Theta_{t-1}\|^2. \tag{78}$$

### D.3.3. EXPRESSING $\widetilde{\Theta}_t$

Since $\widetilde{\Theta}_t$ is a convex combination of $\Theta_t$ and $\Theta_{t-1}$:

$$\widetilde{\Theta}_t = \lambda_t \Theta_t + (1 - \lambda_t)\Theta_{t-1}, \tag{79}$$

we have:

$$\widetilde{\Theta}_t - \Theta_{t-1} = \lambda_t(\Theta_t - \Theta_{t-1}), \|\widetilde{\Theta}_t - \Theta_{t-1}\|^2 = \lambda_t^2 \|\Theta_t - \Theta_{t-1}\|^2. \tag{80}$$

Substituting back into the KL divergence:

$$D_{\mathrm{KL}}(Q \,\|\, P) = \frac{\lambda_t^2}{2\sigma^2} \|\Theta_t - \Theta_{t-1}\|^2. \tag{81}$$

### D.3.4. EFFECT OF CHECKPOINT MERGING ON THE KL DIVERGENCE

Compared to directly using $\Theta_t$ without merging, the divergence would be:

$$D_{\mathrm{KL}}(Q_{\Theta_t} \,\|\, P) = \frac{1}{2\sigma^2} \|\Theta_t - \Theta_{t-1}\|^2. \tag{82}$$

Since $\lambda_t \leq 1$, it follows that:

$$D_{\mathrm{KL}}(Q \,\|\, P) = \lambda_t^2 D_{\mathrm{KL}}(Q_{\Theta_t} \,\|\, P) \leq D_{\mathrm{KL}}(Q_{\Theta_t} \,\|\, P). \tag{83}$$

Thus, checkpoint merging reduces the KL divergence between the posterior and the prior, leading to a tighter generalization bound.

## D.4. Combining Performance and Generalization

We aim to connect the empirical performance and the generalization bound to obtain an overall bound on the expected performance of the merged model on unseen data.

### D.4.1. RELATION BETWEEN PERFORMANCE AND LOSS

Assuming that higher performance corresponds to lower loss (e.g., accuracy is inversely related to error rate), we can define the performance metric $f(\Theta)$ as:

$$f(\Theta) = f_{\max} - L(\Theta), \tag{84}$$

where $f_{\max}$ is the maximum attainable performance, and $L(\Theta)$ is the generalization loss.

### D.4.2. EXPECTED PERFORMANCE ON TEST DATA

Let:

$$f_{\text{test}}(Q) = \mathbb{E}_{\Theta \sim Q}[f_{\text{test}}(\Theta)] = f_{\max} - \mathcal{L}_{\mathcal{D}}(Q), \tag{85}$$
$$f_{\text{train}}(Q) = \mathbb{E}_{\Theta \sim Q}[f_{\text{train}}(\Theta)] = f_{\max} - \mathcal{L}_S(Q). \tag{86}$$

D.4.3. DERIVING THE COMBINED BOUND

From the PAC-Bayesian bound (63), we have:

$$\mathcal{L}_{\mathcal{D}}(Q) \le \mathcal{L}_S(Q) + \epsilon, \tag{87}$$

where:

$$\epsilon = \sqrt{\frac{D_{\mathrm{KL}}(Q \,\|\, P) + \ln\left(\frac{2\sqrt{n}}{\delta}\right)}{2n}}. \tag{88}$$

Substituting into (85):

$$\begin{aligned}
f_{\text{test}}(Q) &= f_{\max} - \mathcal{L}_{\mathcal{D}}(Q) \\
&\ge f_{\max} - (\mathcal{L}_S(Q) + \epsilon) \\
&= (f_{\max} - \mathcal{L}_S(Q)) - \epsilon \\
&= f_{\text{train}}(Q) - \epsilon.
\end{aligned} \tag{89}$$

Thus, we have:

$$f_{\text{test}}(Q) \ge f_{\text{train}}(Q) - \sqrt{\frac{D_{\mathrm{KL}}(Q \,\|\, P) + \ln\left(\frac{2\sqrt{n}}{\delta}\right)}{2n}}. \tag{90}$$

# E. Related Work

**Model Merging in LLM:** Model merging focuses on the unification of several models into one coherent entity, aiming to harness the collective strengths and mitigate the individual weaknesses of each model (Jolicoeur-Martineau et al., 2023), and has recently emerged as a significant trend in the research of Large Language Models. In detail, (Wortsman et al., 2022) proposes model soup to improve accuracy without increasing inference time by averaging weights of multiple fine-tuned models. (Jin et al., 2022; Yu et al., 2024; Wan et al., 2024) investigate the problem of merging individual LM fine-tuned on different datasets to obtain a single model that performs well both across all dataset domains or obtain new capabilities. (Ramé et al., 2024) proposes using model merging to obtain a reliable and robust reward model in RLHF. However, we find that conducting model merging during pretraining receives little attention.

**Bayesian Optimization in NLP:** Bayesian Optimization (BayesOpt) can efficiently optimize objective functions that take a long time to evaluate and are widely applied in NLP. In detail, (Yogatama et al., 2015) leverage BayesOpt for Text Representations. (Ruder & Plank, 2017) learn data selection measures using BayesOpt in transfer learning for sentiment analysis and parsing. (Simpson et al., 2020) proposes using BayesOpt for community QA and summarization, demonstrating its superiority in tasks requiring nuanced feedback interpretation. Besides, (Brochu et al., 2010; Liaw et al., 2018) find that Gaussian process preference learning enables rapid, efficient inference, making it suitable for interactive applications requiring quick user feedback processing. In this paper, unlike previous work, we use BayesOpt to obtain the merging weight for checkpoint merging in LLM pretraining.

# F. Performance on DeepSeek

In our initial experiments, we delve into the possibility of boosting performance through strategic checkpoint merging within the Baichuan pre-trained model, focusing on the checkpoint pairs 1540B and 1760B, along with 2200B and 2420B, in the advanced stages of pretraining. This exploration

| Dataset | Chekpoint-220B | Checkpoint-440B | Uniform Soup | Greedy Soup | Fisher | RegMean |
|---------|---------------|-----------------|--------------|-------------|--------|---------|
| C-Eval | 23.89 | 34.12 | 24.10 | 34.12 | 26.68 | 26.37 |
| CMMLU | 25.54 | 37.11 | 25.78 | 37.11 | 27.46 | 25.23 |
| MMLU | 23.85 | 33.29 | 23.1 | 33.29 | 23.15 | 23.33 |
| GSM8K | 6.82 | 9.10 | 8.04 | 9.10 | 8.16 | 5.46 |

Table 6: The results of merging baichuan2-220B with baichuan2-440B across various benchmark datasets.

| WinoGrand (5-shot) | Pythia 70M | Pythia 410M | Pythia 1.4B | Pythia 2.8B | Pythia 6.9B |
|--------------------|-----------|-------------|-------------|-------------|-------------|
| Training step-142000 | 52.25 | 53.51 | 57.77 | 60.62 | 63.85 |
| Training step-143000 | 51.07 | 53.51 | 57.38 | 60.93 | 63.61 |
| Unifrom Soup | 51.07 | 53.83 | 56.99 | 61.09 | 63.61 |
| Greedy Soup | 51.07 | 53.83 | 57.06 | 60.85 | 63.77 |
| Fisher | 52.08 | 53.88 | 57.96 | 60.57 | 63.84 |
| RegMean | 51.97 | 53.76 | 58.03 | 60.89 | 63.55 |
| **Ours** | **52.35 (+0.10)** | **53.96 (+0.08)** | **58.72 (+0.69)** | **62.04 (+0.95)** | **64.46 (+0.61)** |

Table 7: The results of merging various parameter sizes of Pythia models using different merging methods on the WinoGrande datasets.

aims to uncover patterns of improvement that could be applied across different models. However, our analysis has not extended to other models like DeepSeek. To bridge this gap, we undertake a detailed examination of DeepSeek by evaluating the merging of checkpoints 1800B and 2000B, assessing 100 merging weight points distributed evenly within the [0, 1] interval. Our objective is to ascertain if the trend of performance enhancement through merging observed in Baichuan is also evident in other pre-trained models, specifically through the lens of weight combination efficacy.

The pivotal question guiding our research is: Do similar patterns of performance improvement emerge in other pre-trained models, such as DeepSeek, when applying strategic checkpoint merging, as observed in the Baichuan model?

Our research confirms this hypothesis, revealing that: Specifically, an impressive 70% of the tested merging weight combinations for DeepSeek's checkpoints 1800B and 2000B result in performance enhancements.

This finding indicates that the strategy of merging checkpoints to enhance performance is not unique to the Baichuan model but is also applicable to other pre-trained models like DeepSeek. The consistency of this pattern across different models highlights the potential of checkpoint merging as a universally effective method for optimizing pre-trained model performance during their later training phases.

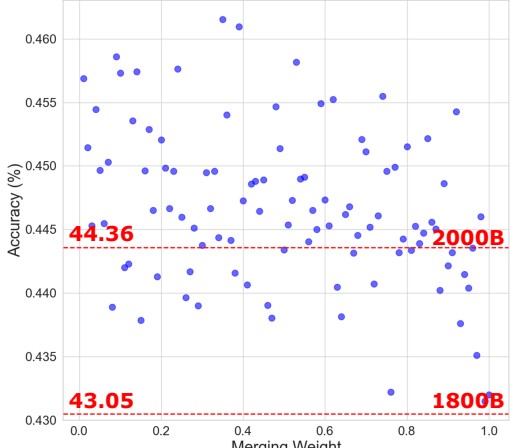

Figure 6: **Validation of checkpoint merging effectiveness on DeepSeek models:** merging DeepSeek-1800B and DeepSeek-2000B checkpoints with uniformly sampled weights from [0,1] on the C-Eval dataset. The performance curve demonstrates that **69%** of the tested merging weight combinations result in performance enhancements beyond the superior base model (DeepSeek-2000B).

## G. All the results of baichuan2-220B with baichuan2-440B across various benchmark datasets.

As outlined in our pilot experiments, performing weight merging on checkpoints of LLMs during the early stages of pre-training leads to a degradation in performance. In the Table 6, we present our experimental results on weight merging between baichuan2-220B and baichuan2-440B models. We use Uniform Soup, Greedy Soup, Fisher, and RegMean to conduct merging and measure the post merging performance. Without exception, none of the outcomes show performance surpassing that of the superior base model (baichuan2-440B). This evidence suggests that conducting checkpoint merging in the early pre-training phase is not a viable strategy.

| SciQ(5-shot) | Pythia70M | Pythia410M | Pythia1.4B | Pythia2.8B | Pythia6.9B |
|---|---|---|---|---|---|
| Training step-142000 | 56.30 | 88.70 | 92.50 | 94.10 | 94.60 |
| Training step-143000 | 58.20 | **89.40** | 92.30 | 93.60 | 94.90 |
| Unifrom Soup | 57.40 | 89.10 | 92.10 | 94.60 | 94.60 |
| Greedy Soup | 57.40 | 89.10 | 92.50 | 94.10 | 95.00 |
| Fisher | 58.10 | 88.70 | 92.90 | 93.90 | 94.50 |
| RegMean | 57.40 | 88.90 | 92.70 | 94.00 | 94.60 |
| **Ours** | **58.30 (+0.10)** | **89.20 (-0.20)** | **93.10 (+0.20)** | **94.80 (+0.70)** | **95.30 (+0.30)** |

Table 8: The results of merging various parameter sizes of Pythia models using different merging methods on the SciQ datasets.

| ARC-Easy(5-shot) | Pythia70M | Pythia410M | Pythia1.4B | Pythia2.8B | Pythia6.9B |
|---|---|---|---|---|---|
| Training step-142000 | 35.86 | 54.97 | 62.88 | 66.79 | 69.74 |
| Training step-143000 | 36.57 | 54.76 | 63.01 | 67.00 | 69.87 |
| Unifrom Soup | **36.70** | 54.67 | 63.05 | 67.00 | 70.12 |
| Greedy Soup | **36.70** | 54.67 | 62.96 | 67.17 | 69.95 |
| Fisher | 36.27 | 55.10 | 63.04 | 66.87 | 69.89 |
| RegMean | 36.14 | 55.21 | 62.98 | 67.01 | 70.22 |
| **Ours** | **36.58 (-0.12)** | **55.77 (+0.56)** | **63.13 (+0.08)** | **68.05 (+0.88)** | **70.82 (+0.60)** |

Table 9: The results of merging various parameter sizes of Pythia models using different merging methods on the ARC-Easy datasets.

| Baichuan2-1980B&2200B | EI | MPI | LCB | GP-hedge |
|---|---|---|---|---|
| C-Eval(5-shot) | 56.69 | 56.53 | 56.14 | **56.73(+0.04)** |
| CMMLU(5-shot) | 56.88 | 56.89 | 56.50 | **57.05(+0.16)** |
| MMLU(5-shot) | 54.60 | 54.54 | 54.62 | **54.77(+0.15)** |
| GSM8K(4-shot) | 22.07 | 21.98 | 21.76 | **22.17(+0.10)** |
| Average | 47.56 | 47.49 | 47.26 | **47.68(+0.12)** |

Table 10: The results of the ablation study for different acquisition strategies when merging Baichuan2-1980B with Baichuan2-2200B.

# H. The table of different model size on Varying datasets

# I. Acquisition Functions

We conduct an ablation study on different acquisition functions and find that using GP-hedge yields better results compared to individual acquisition functions. For instance, in the C-Eval (5-shot) task, GP-hedge achieves the highest performance with a score of 56.73%, which is an improvement of 0.04% over the best individual acquisition function, EI, which scores 56.69%. Similarly, in the CMMLU (5-shot) task, GP-hedge outperforms the others with a score of 57.05%, marking an improvement of 0.16% over the best individual acquisition function, MPI, which scores 56.89%. For the MMLU (5-shot) task, GP-hedge again shows superior performance with a score of 54.77%, surpassing the best individual function, LCB, by 0.15%. Lastly, in the GSM8K (4-shot) task, GP-hedge scores 22.17%, an improvement of 0.10% over EI, which scores 22.07%. On average, GP-hedge shows a significant improvement with an average score of 47.68%, which is 0.12% higher than the best individual acquisition function, EI, which averages 47.56%.

# J. Pilot Experiments performance on CMMLU Dataset

In this section, we describe the performance of merging checkpoints on the CMMLU dataset.

In accordance with the research question outlined in pilot experiments, we extended our analysis to the CMMLU dataset to corroborate the findings from the C-Eval dataset. Utilizing the same methodology, we evaluated all possible pairwise merging, totaling 55 combinations ($C_{11}^2$). Employing the greedy soup strategy as outlined by (Wortsman et al., 2022), checkpoints are sequentially incorporated into the soup if they demonstrate an improvement in accuracy on the development data.

The analysis yields observations that are consistent with those obtained from the C-Eval dataset, reinforcing the generality of our findings. Specifically, we observe that:

(1) Adjacent checkpoint merging yields superior performance: Echoing the C-Eval dataset's findings, merging two checkpoints from consecutive training phases in the CMMLU dataset generally led to performance enhancements over individual checkpoints. Notably, merging Baichuan2-1980B with Baichuan2-2200B achieved an accuracy of 56.57% on the CMMLU dataset, surpassing the 56.29% accuracy of Baichuan2-2200B when assessed independently. This not only highlights the effectiveness of adjacent checkpoint merging but also indicates a significant improvement over the final checkpoint, Baichuan-2420B (with an accuracy of 56.78%), by elevating the test accuracy by 0.10%.

(2) Merging distant checkpoints leads to performance deterioration: In line with the C-Eval dataset observations, merging significantly disparate checkpoints, such as Baichuan2-220B with Baichuan2-2200B, resulted in a notable performance decline, with accuracy dropping to 25.26% on the CMMLU dataset. This outcome closely mirrors the performance of the lesser-trained checkpoint, Baichuan2-220B, which has an accuracy of 25.54%, underscoring the negative impact of merging widely separated checkpoints.

The CMMLU dataset findings reinforce the C-Eval dataset's results, highlighting the critical importance of strategic checkpoint merging for enhanced model performance across diverse datasets.

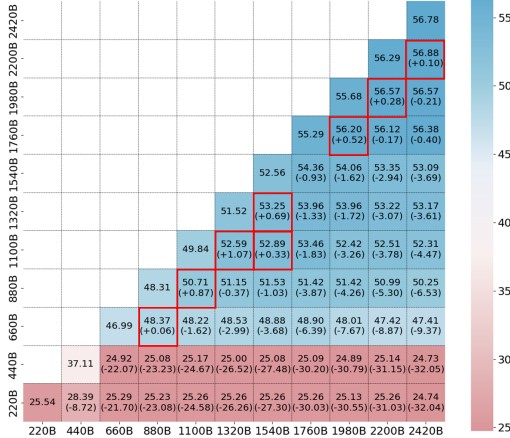

Figure 7: **Performance comparison of pairwise checkpoint merging using the Greedy Soup method on the CMMLU benchmark across all possible combinations of 11 Baichuan2 checkpoints.** This heatmap visualization corroborates findings from the C-Eval dataset, showing that adjacent checkpoint merging (diagonal region) consistently yields performance improvements, while distant checkpoint combinations result in substantial performance degradation.

