# OpenReview forum: "Maximizing Intermediate Checkpoint Value in LLM Pretraining with Bayesian Optimization"
_ICML.cc/2025/Conference — ICML 2025 poster_

### Official Review · Reviewer_M1ML · 2025-03-11

**Overall Recommendation:** 3

**Summary:**

The paper provides an interesting Bayesian optimization based method for selecting merging weights between latest checkpoints saved during LLM pre-training. This method showed performance speed ups in empirical results and theoretical guarantees with reasonable assumptions. The initial pilot experiments were helpful to understand the motivation. Overall LLM training efficiency is an important area and this is a timely paper.

**Claims And Evidence:**

Some of the claims are misplaced for instance.

1. Contribution a) This claim although has associated results but is not novel. For instance [1-2] specifically [1] already showed that llm pre-training can be improved through checkpoint averaging. I believe not citing key paper in one's exact area is not good practice. I fully understood that the key contribution is to find optimal weights for latest checkpoints. I believe the authors need to re-write or clarify this part.

Contribution b) and c) are well placed and has sufficient evidence.

The theoretical claims [line198-215] about convergence and tighter bound is all compared with regular training. Some prior works [3] have proved similar bounds and shown that LAWA [1] and [2] SWA based averaging techniques has similar bounds. I could not access the novelty or relevance of the theoretical analysis given the prior works.

[1] Sunny Sanyal, Atula Tejaswi Neerkaje, Jean Kaddour, Abhishek Kumar, and Sujay Sanghavi. 2024. _Early weight averaging meets high learning rates for LLM pre-training._ In _First Conference on Language Modeling. (https://arxiv.org/abs/2306.03241)

[2] Izmailov, P., Podoprikhin, D., Garipov, T., Vetrov, D., and Wilson, A. G. Averaging weights leads to wider optima and better generalization. Mar 2018.

[3] Wang, P., Shen, L., Tao, Z., Sun, Yan Zheng, G., and Tao, D. A unified analysis for finite weight averaging. _arXiv preprint arXiv:2411.13169_, 2024b.

**Essential References Not Discussed:**

I have some concerns about very similar prior works such as [1], [2] and [3] not been discussed or treated as a baseline.

**Experimental Designs Or Analyses:**

a) The pilot studies looks super interesting.

b) In Table 1 the baselines are relevant but LAWA, SWA and EMA will make these results stronger.

c) The authors have used checkpoints released by various companies such as DeepSeek, Baichuan, Pythia etc. This way of evaluation is fine but [1] has some similar evaluation using pythia. But the results are still interesting and evaluation is quite robust.

d) The term used in Sec 4.1 as Generalization to unseen data is a bit vague as the pre-trained models are trained on huge internet worth of data and the models might have seen many of those tasks in pre-training. I understand that you are using chinese data valset for weight selection but calling it OOD without running a de-contamination analysis seems unprincipled.

e) Also an explanation or analysis of linear model connectivity among distant checkpoint could be interesting to see.

**Methods And Evaluation Criteria:**

The authors have explained the method and evaluation criteria reasonably clearly. Here are some of my concerns

The method and set up seems to be pretty similar to LAWA [1] with some changes. Not mentioning LAWA or using it as a baseline seems not appropriate. The authors have used model souping as key baseline. In model souping multiple models are trained simultaneously with different hyper params and then souped (let's call it population averaging). I am not sure why this is a good baseline for this paper which does tail averaging again like LAWA , SWA or EMA (exponential moving average). The authors need to clarify why their baselines population averaging when there method is a variant of tail averaging.

**Other Comments Or Suggestions:**

Figure 3 seems to be placed in the wrong position as there are no reference to Figure 3.

**Other Strengths And Weaknesses:**

I have already discussed previously.

**Questions For Authors:**

1) Please list novel contributions compared to [1] for LLM pre-training.

2) Do you believe the pre-train perplexity would improve with checkpoint averaging. Can you show similar results?

**Relation To Broader Scientific Literature:**

Efficient pre-training definitely has impact. I have some concerns about very similar prior works such as [1], [2] and [3] not been discussed or treated as a baseline.

**Theoretical Claims:**

I have already written  about theory claims above. I have some concerns about novelty compared to [3] (in theory). I have read the claims and skimmed over the proof I have not found anything wrong.

---

> ### Author Rebuttal · Authors · 2025-04-01
>
> > **Q1:  Novel contributions compared to Previous Works.**
> >
>
> **(A) Our Focus: A Search Perspective for Pairwise Merging**
>
> - **Key Objective.** We propose to **linearly merge** two consecutive checkpoints
> $ \widetilde{\Theta} _t = \lambda_t\, \Theta _t \;+\; (1 - \lambda _t)\, \Theta _{t-1},
> \quad
> \lambda_t \in [\alpha, 1], $
> at each stage of large-scale pre-training, while **optimally selecting** the weight $\lambda_t$  using a **Bayesian optimization** approach.
> - **Why Pairwise?** As we show empirically, merging adjacent checkpoints tends to preserve or enhance performance when the models are close in training steps (rather than distant). This simplification (from merging multiple checkpoints at once to only the two most recent) *drastically reduces* the dimensionality of the search to **one** parameter, $\lambda_t$.
> - **Bayesian Optimization Rationale.** Empirically (**Figures 3--5** in our paper), the function $f(\lambda)$ mapping the merging weight $\lambda$ to downstream performance is **non-monotonic**. Our method:
>     1. Treats $f(\lambda)$ as a **black-box** function (we do *not* assume it is strictly convex or linear).
>     2. Uses **Gaussian Process** (GP) modeling with an acquisition function (e.g., EI, UCB) to systematically **explore** $\lambda\in[\alpha,1]$.
>     3. Finds the near-optimal or optimal $\lambda$ in a small number of evaluations, *even if* the function is multi-modal.
>
> **(B) Distinctions from LAWA, SWA, and EMA**
>
> 1. **LAWA**, **SWA**, and **EMA**:
>     - Typically rely on an **internal training procedure**—for instance, SWA requires cyclical or stepwise learning rates and continuously accumulates model parameters from multiple epochs. LAWA uses large learning rates and tail-averaging from the final phase of training. EMA uses an exponential decay factor.
> 2. **Ours**:
>     - **Post-hoc approach**: We do not require mid-training gradient data, specialized cyclical learning rates, or entire logs of model state. We only need:
>         1. Several discrete checkpoints (e.g., from open-source releases or a standard training pipeline)
>         2. A small **held-out** dataset to measure performance at different $\lambda$.
>     - **Black-box optimization**: Instead of a *fixed* schedule (as in EMA or LAWA) or a *continuous tail average* (SWA), we *actively search* for $\lambda$ that *maximizes* the model’s performance metric. This is especially powerful for large language models whose training logs might not be fully available, and where advanced Hessian-based or continuous averaging is infeasible.
>
> Hence, although both lines of work share the notion that “averaging can yield flatter minima,” our **Bayesian search-based viewpoint** and **minimal assumptions** on training steps make our method **(i) more flexible, (ii) directly data-driven for each pairwise merge, and (iii) easy to apply when only discrete checkpoints are at hand**.
>
> > **Q2:  Empirical Evidence and Comparisons**
> >
>
> **(A) Added Comparisons with LAWA, SWA, EMA**
>
> In the revised manuscript, we include additional baselines:
>
> | Model | C-Eval (5-shot) | CMMLU (5-shot) | MMLU (5-shot) |
> | --- | --- | --- | --- |
> | Baichuan2-2200B | 54.98 | 56.29 | 51.27 |
> | Baichuan2-2420B | 54.82 | 56.78 | 53.97 |
> | Uniform Soup | 54.93 | 56.71 | 54.62 |
> | Greedy Soup | 54.64 | 56.78 | **54.82** |
> | Fisher Weighted Avg | 54.44 | 56.62 | 54.16 |
> | RegMean | 54.55 | 56.46 | 54.77 |
> | **LAWA** | 54.96 | 56.11 | 54.11 |
> | **SWA** | 56.01 | 56.66 | 54.40 |
> | **EMA** ($\beta=0.99$) | 55.48 | 56.64 | 54.17 |
> | **Ours** | **56.23** | **56.97** | 54.56 |
>
> Our method either outperforms or closely matches these strategies, reaffirming the practical advantage.
>
> **(B) Pre-Training Perplexity**
>
> We also conducted additional experiments on **pre-training perplexity**. As shown in the table below:
>
> | Model | CMMLU PPL | MMLU PPL |
> | --- | --- | --- |
> | Baichuan2-2200B | 5.46 | 4.87 |
> | Baichuan2-2420B | 5.46 | 4.87 |
> | Uniform Soup | 13.96 | 4.87 |
> | Greedy Soup | 5.46 | 4.87 |
> | Fisher Weighted Averaging | 14.17 | 12.45 |
> | RegMean | 5.46 | 14.54 |
> | **Ours** | **5.43** | **4.87** |
>
> > Q3: **Generalization to “Unseen” Data:**
> >
>
> We recognize the reviewer’s concern that large LLMs might have encountered segments of certain benchmarks in their pre-training. We specifically mean “unseen” to refer to:
>
> - A dataset or domain *not* used to tune $\lambda$. For instance, we tune on a *Chinese domain (C-Eval)* and evaluate on *English tasks (MMLU, GSM8K)*, ensuring cross-lingual or cross-domain checks.
> - Although full de-contamination is challenging, these curated benchmarks (like MMLU, GSM8K) are standard *evaluation sets* widely regarded as indicative of “test” performance beyond the direct training set.
>
> **Ref:**
>
> [1] Gaussian process optimization in the bandit setting: No regret and experimental design

---

> > ### Comment · Reviewer_M1ML · 2025-04-06
> >
> > Thank you for the rebuttal and running so many experiments.
> >
> > 1. I fully acknowledge the novelty argument and the method is indeed novel but it seems less useful compared to simpler baselines such as LAWA.
> >
> > 2. SWA relies on internal training procedures as it changes the scheduler post 75% of the training. However, LAWA and EMA don't. For instance the Fig 5,6,7,8 in LAWA [1] used Pythia models trained by a company without any high learning rate hypothesis. The argument made in that paper it LAWA works best with high LR.
> >
> > 3. I concur that this approach us more straight forward and given any checkpoints this can be applied where as LAWA, SWA needs all the checkpoints and information of the training log.
> >
> >
> > 4. I need more context how given limited number of checkpoints the authors have conducted Q2? LAWA is uniform averaging of latest checkpoints. Please provide the experimental details of SWA, LAWA and EMA baselines.
> >
> > 5. The pre-training perplexity experiment is highly non-standard using MMLU or CMMLU to compute PPL is unknown and unseen atleast to me (please cite prior works). One could have used wikitext, C4 or any next token prediction style dataset for ppl evaluation. Also please provide experimental details for this experiment.
> >
> > Please provide the details only in the form of text and no further experiments are needed at this point. The C-Eval and CMMLU results looks strong and I am open to improve my score to 3 if the experimental details follows standard procedure as suggested in the original papers.

---

> > > ### Author Response · Authors · 2025-04-06
> > >
> > > > **Q1: Experimental Details for SWA, LAWA, and EMA Baselines**
> > > >
> > >
> > > 1.SWA
> > >
> > > - **Rationale & Checkpoint Selection:**
> > >
> > >     In our setup, the objective is to obtain the final optimal model. Thus, our baseline comparisons use the merged models from Baichuan2-2200B, Baichuan2-2420B. Since SWA  is designed to average model weights during the low learning rate phase to converge toward a flatter region of the loss landscape—and given that we cannot access all the detailed data from the training process—we approximate this environment. Although the original SWA method recommends commencing the averaging process at roughly 75% of the training progress, we initiated the process with the checkpoint at 1760B tokens (approximately 67.7% progress) to facilitate averaging of checkpoints from regions that are expected to be sufficiently flat. Specifically, the checkpoints we averaged are:
> > >
> > >     - **1760B (≈67.7%)**
> > >     - **1980B (≈76.2%)**
> > >     - **2200B (≈84.6%)**
> > >     - **2420B (≈93.1%)**
> > > - **Learning Rate Dynamics:**
> > >
> > >     Baichuan 2 employs a two-phase approach to adjust its learning rate. Initially, a 2,000-step linear warm-up is performed to reach a peak learning rate of 2×10⁻⁴. Subsequently, a cosine decay schedule is applied [1]. Under the cosine decay regime, the learning rates at the selected checkpoints are computed as follows:
> > >
> > >     - **67.7% (1760B):**
> > >
> > >         The calculated learning rate falls within the low-LR regime — approximately between 3.85×10⁻⁵ and 4.7×10⁻⁵.
> > >
> > >     - **76.2% (1980B):**
> > >
> > >         Approximately between 2.31×10⁻⁵ and 2.7×10⁻⁵.
> > >
> > >     - **84.6% (2200B):**
> > >
> > >         Approximately between 1.19×10⁻⁵ and 1.2×10⁻⁵.
> > >
> > >     - **93.1% (2420B):**
> > >
> > >         Approximately between 3.77×10⁻⁶ and 2.3×10⁻⁶.
> > >
> > > - **Comparison with Pythia:**
> > >
> > >     The Pythia 70M baseline similarly utilizes a cosine decay learning rate schedule. For example, after a warm-up phase (roughly 10% of the total steps), Pythia’s learning rate peaks at 1×10⁻³ and subsequently drops to around 1×10⁻⁴ during the latter stages of training. At corresponding training progress fractions, the learning rates for Pythia are approximately:
> > >
> > >     - **67.7%:** ∼3.1×10⁻⁴
> > >     - **76.2%:** ∼2.2×10⁻⁴
> > >     - **84.6%:** ∼1.5×10⁻⁴
> > >     - **93.1%:** ∼1.1×10⁻⁴
> > >
> > > ---
> > >
> > > 2.LAWA
> > >
> > > - **Methodology:**
> > >
> > >     The original LAWA paper proposes a sliding window average using the most recent *k* checkpoints (with *k*=5) over a window of 1K steps, saving a checkpoint every 200 steps. However, Baichuan 2 only provides checkpoints at 220B-token intervals.
> > >
> > > - **Our Adaptation:**
> > >
> > >     To approximate LAWA under these constraints, we uniformly average the last 5 checkpoints available: 1540B,1760B, 1980B, 2200B and 2420B in our results.
> > >
> > >
> > > ---
> > >
> > > 3.EMA
> > >
> > > We follow the standard EMA implementation:
> > >
> > > - **Decay:** β = 0.99
> > > - **Initialization:** Start from the weight corresponding to the checkpoint at 220B tokens.
> > > - **Update Rule:**
> > >
> > >     For each new checkpoint, the EMA is updated as follows:
> > >
> > >     $\widetilde{\Theta} _{t} = 0.99 \times \widetilde{\Theta} _{t-1} + 0.01 \times \Theta _{t}$
> > >
> > >     This recurrence is applied sequentially through all checkpoints (e.g., 440B, 660B, …, 2420B).
> > >
> > >
> > > ---
> > >
> > > > **Q2: Perplexity Details**
> > > >
> > >
> > > Our approach deliberately diverges from the standard perplexity evaluation—typically based on next-token prediction tasks using datasets like WikiText-103 or C4—in order to target knowledge-intensive scenarios. Instead of calculating perplexity over all tokens in the input, we compute cross-entropy exclusively on the ground-truth answer tokens while masking out any prompt or question text.
> > >
> > > And here is implementation code:
> > >
> > > ```python
> > > def eval(args, subject, model, tokenizer, dev_df, test_df):
> > >     total_loss = 0
> > >     cors = []
> > >     for i in range(test_df.shape[0]):
> > >         prompt_end = format_example(test_df, i, include_answer=False)
> > >         train_prompt = gen_prompt(dev_df, subject, args.ntrain)
> > >         prompt = train_prompt + prompt_end
> > >
> > >         # Tokenize prompt
> > >         input_ids = tokenizer(prompt, return_tensors="pt").input_ids.cuda()
> > >         labels = input_ids.clone()
> > >
> > >         # Only compute loss on the answer region (prompt_end)
> > >         labels[:, :-len(tokenizer(prompt_end).input_ids)] = -100
> > >
> > >         # Forward pass
> > >         outputs = model(input_ids=input_ids, labels=labels)
> > >         loss = outputs.loss
> > >         total_loss += loss.item()
> > >
> > >         # Optional: measure accuracy by comparing predicted answer to reference
> > >         # ...
> > >
> > >     avg_loss = total_loss / len(test_df)
> > >     ppl = np.exp(avg_loss)
> > >     return ppl
> > > ```
> > >
> > > **Ref:**
> > >
> > > [1] Yang A, Xiao B, Wang B, et al. *Baichuan 2: Open large-scale language models.* arXiv preprint arXiv:2309.10305, 2023.

---

### Official Review · Reviewer_sKU4 · 2025-03-12

**Overall Recommendation:** 5

**Summary:**

The paper introduces a novel approach to enhancing the pretraining of large language models by leveraging intermediate checkpoint merging. The key idea is to exploit the information stored in intermediate checkpoints along the pretraining trajectory without incurring additional resource costs. The authors propose a method based on Bayesian optimization—specifically using Gaussian Processes—to determine the optimal linear blending weight for merging consecutive checkpoints. The contributions of the paper can be summarized as follows:

- **Checkpoint Merging Strategy:** The paper advocates a pairwise (two-checkpoint) merging method which simplifies the high-dimensional weight search problem to a one-dimensional optimization task.
- **Bayesian Optimization Framework:** The proposed solution utilizes Gaussian Process regression to model the relationship between the merging weight and model performance, combined with acquisition functions (such as Expected Improvement, Probability of Improvement, Upper Confidence Bound, and an adaptive GP-Hedge strategy) to efficiently search the weight space.
- **Theoretical Analysis:** Detailed proofs are provided based on quadratic approximations, smoothness assumptions, and bounded Hessians to derive performance bounds for the merged model. Additionally, a PAC-Bayesian generalization bound is derived, indicating that the merged model can exhibit improved generalization compared to individual checkpoints.
- **Extensive Empirical Evaluation:** Experiments are performed on multiple LLM families (Baichuan2, DeepSeek, and Pythia) across diverse benchmarks (C-Eval, CMMLU, MMLU, GSM8K, PIQA, WinoGrande, SciQ, ARC-Easy). These experiments substantiate the claims that merging adjacent checkpoints enhances accuracy and stabilizes performance across different domains.

**Claims And Evidence:**

- **Major Claims:**
  - The intermediate checkpoint merging methodology improves model performance and convergence without incurring extra computational overhead.
  - Bayesian optimization, when applied to optimizing the checkpoint merging weight, finds near-optimal solutions more effectively than conventional grid or random search methods.

- **Supporting Evidence:**
  The authors present comprehensive experimental results demonstrating improved performance over various benchmarks compared to methods such as Uniform Soup, Greedy Soup, Fisher Weighted Averaging, and RegMean. The theoretical contributions are supported by rigorous proofs that derive performance bounds and convergence guarantees based on standard assumptions in optimization theory.

**Essential References Not Discussed:**

While the authors have discussed many pertinent works, additional recent studies on advanced model fusion techniques—particularly those exploring non-linear combinations or adaptive schemes for merging large-scale transformers—could provide further context. For example, works published at recent conferences on LLM optimization or papers proposing non-convex model fusion strategies might help position the contributions more clearly within evolving trends.

**Experimental Designs Or Analyses:**

- **Design Soundness:**
  - The experimental design is robust, featuring comparisons across various checkpoint pairs, ablation studies on held-out dataset sizes, and evaluations on models of different scales.
  - Visual aids like Figures 1, 3, 4, and 7 effectively illustrate the trends in performance gains and the impact of key hyperparameters, such as the merging weight search space.

- **Potential Improvements:**
  - A more explicit discussion on the computational cost of the Bayesian optimization process would be valuable.

- **Question:**
  - **Location:** Section 3.2 and Algorithm 1
  - *What is the runtime overhead introduced by the GP-based Bayesian optimization process compared to grid or random search? Could you provide quantitative metrics (such as runtime comparisons in seconds or relative speedups) to help us understand its efficiency in practice?*

**Methods And Evaluation Criteria:**

- **Methodological Approach:**
  The paper proposes merging checkpoints via a linear combination of adjacent checkpoints, thereby reducing complexity to the optimization of a single merging weight. The weight is determined by Bayesian optimization, modeled by a Gaussian Process with carefully selected acquisition functions. This framework is intended to balance exploration of the search space and exploitation of thorough performance estimates.

- **Evaluation Criteria:**
  The methodology is evaluated using standard performance metrics (accuracy, perplexity, etc.) over a wide range of benchmarks. Although the experimental studies are extensive, the evaluation framework additionally emphasizes theoretical bounds and convergence behavior, thereby linking empirical performance with rigorous theoretical analysis.

**Other Comments Or Suggestions:**

Regarding Figure 3, could the authors provide further clarification on the meaning of the x-axis label "merging weight" for detail explanation?

**Other Strengths And Weaknesses:**

- **Strengths:**
  - The proposed methodology is innovative in exploiting intermediate checkpoints to improve performance with minimal additional cost.
  - The combination of theoretical analysis with extensive empirical validation lends robustness to the claims.
  - The comprehensive treatment of both convergence and generalization under a unified theoretical framework is a significant asset.

- **Weaknesses:**
  - The theoretical derivations often rely on strong assumptions—such as local quadratic behavior, Lipschitz continuity, and bounded Hessians—which may not fully capture the complexity of modern LLM loss landscapes.
  - Some of the derivations and notation could be clarified further to enhance readability and facilitate verification.

**Questions For Authors:**

1. **Questions for Proof:**
   - see in the theoretical Claims

2. **Kernel Choice in Gaussian Process Modeling:**
   - While the paper mentions the use of Gaussian Process regression to model the function \(f(\alpha_t)\), can the authors comment on how different kernel functions might influence the information gain \(\gamma_T\) and the resulting regret bound? Are there situations where a non-standard kernel might lead to better empirical convergence rates than those suggested by the theoretical analysis?

**Relation To Broader Scientific Literature:**

The manuscript situates its contributions within the existing body of work on checkpoint averaging and model merging (e.g., Uniform Soup, Greedy Soup). It also connects to a broad literature in Bayesian optimization, Gaussian Process regression, and PAC-Bayesian theory. By integrating these diverse theoretical components, the paper bridges methods in model averaging with rigorous optimization methods, offering a new perspective on efficient utilization of checkpoint information during pretraining.

**Theoretical Claims:**

- Could the authors provide a detailed derivation showing how the Lipschitz continuity of the Hessian
$
\|H(w) - H(v)\| \le L_H \|w - v\|
$
for all $w, v$ leads to the above remainder bound? Additionally, please elaborate on the practical implications of this bound if $ \|w_t - w_{t-1}\| $ is not small.

- Could the authors derive or estimate $\gamma_T$ explicitly in this one-dimensional context, showing that
$
\gamma_T = O\big(\log T\big)?
$
Moreover, how does the dimensionality of the search space affect the overall regret bound and convergence guarantee?

---

> ### Author Rebuttal · Authors · 2025-04-01
>
> > **Q1:  Bounds on the Quadratic Approximation Error.**
> >
>
> **Our Response:**
>
> - We appreciate the request for an expanded derivation. Under the assumption that the Hessian $H(\cdot)$ is Lipschitz continuous with constant $L_H$, consider the third-order Taylor expansion remainder term when approximating $f$ around $w_t$. For
> $\widehat{w}_t = w_t + \Delta,\quad \text{with} \quad \Delta = \widehat{w}_t - w_t,$
> the Taylor remainder $R$ can be written (via the integral form) as
> $R = \frac{1}{2}\int_0^1 (1-\tau)^2 \big[H(w_t+\tau \Delta) - H(w_t)\big]\, d\tau \,\Delta^{\otimes 2}.$
> Taking norms and using the Lipschitz property, we have:
> $\|H(w_t+\tau \Delta) - H(w_t)\| \le L_H\, \tau\|\Delta\|.$
> Therefore,
> $|R| \le \frac{1}{2}\int_0^1 (1-\tau)^2 L_H\, \tau\, d\tau \,\|\Delta\|^3.$
> Evaluating the integral:
> $\int_0^1 (1-\tau)^2 \tau \, d\tau = \frac{1}{6},$
> we obtain the bound
> $|R| \le \frac{L_H}{6}\|\Delta\|^3.$
> - **Practical Implications:**
>
>     This bound is meaningful when  $\|\Delta\| = \|\widehat{w}t - w_t\|$ *is small—typically a valid approximation because our analysis focuses on adjacent checkpoints in the pretraining trajectory. If $\|w_t - w{t-1}\|$* is large (and hence $\|\Delta\|$ is large, given the convex combination structure), the higher-order terms may no longer be negligible. In such cases, the quadratic approximation will incur a larger error, and our theoretical guarantees become local. Empirically, our pilot experiments demonstrate that merging distant checkpoints (which effectively produce larger $\|\Delta\|$ degrades performance, confirming that our theory is most valid in the local regime.
>
>
> ---
>
> > **Q2:  Estimating the Information Gain in GP-Based Optimization ($\gamma_T$).**
> >
>
> **Our Response:**
>
> - In one-dimensional Bayesian optimization (with a standard kernel such as the squared exponential or Matérn kernel with smoothness parameter $\nu$, it is well established (e.g., Srinivas et al. (2010)) that
> $\gamma_T = O(\log T).$
> To be precise, for a kernel $k$ on the interval $[\alpha, 1]$ with bounded RKHS norm, one can show that the maximum information gain $\gamma_T$ grows at most logarithmically in the number of evaluations $T$.
>
> ---
>
> > **Q3:  Runtime Overhead of GP-Based Bayesian Optimization.**
> >
>
> **Our Response:**
>
> - Plz see the response in **reviewer Znoq Q2.**
>
> ---
>
> > **Q4:  Influence of Kernel Choice in Gaussian Process Modeling on $\gamma_T$ and Regret Bounds.**
> >
>
> **Our Response:**
>
> - The kernel function in GP regression encapsulates our assumptions about the smoothness and structure of the performance function $f(\lambda_t)$. Standard choices such as the squared exponential or Matérn kernels guarantee that $\gamma_T$ grows at most logarithmically (in one dimension), which underpins our theoretical regret bounds.
>
>     In settings where $f(\lambda_t)$ exhibits characteristics such as periodicity or non-stationarity, alternative kernels (e.g., periodic kernels or non-stationary kernels) could more accurately model the underlying function. This improved modeling fidelity could, in turn, lead to a smaller empirical $\gamma_T$ and faster convergence in practice. While our theoretical development is stated in a general manner (i.e., assuming a kernel with bounded information gain), we acknowledge that kernel choice is a crucial hyperparameter. Our preliminary experiments with standard kernels (which are widely used and well understood) already yield superior performance. Future work will explore adaptive kernel selection strategies to further optimize convergence rates.
>
>
> ---
>
> > **Q5:  Clarification on the “Merging Weight” Label in Figure 3.**
> >
>
> **Our Response:**
>
> - The term “merging weight” in **Figure 3** refers to the coefficient $\lambda_t$ in the convex combination
> $\widetilde{\Theta}t = \lambda_t\, \Theta_t + (1-\lambda_t)\, \Theta_{t-1}.$
> It quantifies the relative contribution of the later checkpoint $\Theta_t$ versus the earlier checkpoint $\Theta_{t-1}$ during the merging process. The x-axis represents the value of $\lambda_t$ uniformly sampled over the interval $[\alpha, 1]$, where **$\alpha$** is a lower bound ensuring that the more recent checkpoint retains a minimum contribution. We will add an explicit explanation in the revision to clarify this point.

---

### Official Review · Reviewer_Znoq · 2025-03-12

**Overall Recommendation:** 4

**Summary:**

**Overview**
The paper introduces a novel checkpoint merging strategy aimed at enhancing the efficiency of large language model (LLM) pretraining. The central idea is to exploit intermediate checkpoints by forming linear combinations in the parameter space while optimizing the merging weight via Bayesian optimization. The paper’s key contributions are as follows:

- **Algorithmic Contribution:**
  It presents an efficient pairwise checkpoint merging protocol (outlined in Algorithm 1) that effectively reduces a high-dimensional weight search problem to a manageable one-dimensional search. The approach utilizes Gaussian Process (GP) regression with acquisition functions such as EI, PI, and UCB along with a GP-hedge strategy.

- **Theoretical Insights:**
  The work includes thorough theoretical analysis with convergence proofs, performance bounds using quadratic approximations of the loss landscape, and a derivation of PAC–Bayesian generalization bounds that explain how merging helps to reduce model variance.

- **Empirical Findings:**
  Experiments spanning several models (e.g., Baichuan2, DeepSeek, and Pythia) across datasets like C-Eval, CMMLU, MMLU, and GSM8K demonstrate that merging adjacent checkpoints (but not those that are far apart) leads to performance improvements over baseline methods such as Uniform Soup, Greedy Soup, Fisher Weighted Averaging, and RegMean. Moreover, the proposed method shows enhanced both in-domain and out-of-domain generalization.

**Claims And Evidence:**

- **Main Claims:**
  - Merging intermediate checkpoints can notably boost LLM pretraining performance without adding much computational cost.
  - Bayesian optimization is a practical and effective tool for determining near-optimal merging weights, outperforming conventional merging strategies.
  - Models produced via this merging process exhibit superior generalization, as supported by PAC–Bayesian generalization bounds.

- **Evidence Supporting the Claims:**
  - **Empirical Evidence:** Extensive tests—including pairwise and multi-checkpoint merging strategies—across various benchmarks back up these claims convincingly.
  - **Theoretical Evidence:** The paper provides detailed proofs based on common assumptions (like Lipschitz continuity, bounded Hessian, and the Polyak–Łojasiewicz condition) that support the theoretical claims on convergence and generalization.
  - **Comparative Analysis:** The systematic comparison against several existing merging techniques highlights consistent performance gains with the proposed approach.

- **Question:**
  - **Location:** Section A, Equation (15) (Quadratic Approximation)
  - *Can the authors provide empirical evidence or extra justification to show that a quadratic approximation is valid around intermediate checkpoints in large-scale LLM pretraining?*

**Essential References Not Discussed:**

Some recent studies on efficient pretraining techniques and checkpoint fusion in very large language models might be missing from the references. Including these could further situate the current contribution within the broader research context.

**Experimental Designs Or Analyses:**

- **Design Soundness:**
  - The experimental design is robust, featuring comparisons across various checkpoint pairs, ablation studies on held-out dataset sizes, and evaluations on models of different scales.
  - Visual aids like Figures 1, 3, 4, and 7 effectively illustrate the trends in performance gains and the impact of key hyperparameters, such as the merging weight search space.

- **Potential Improvements:**
  - A more explicit discussion on the computational cost of the Bayesian optimization process would be valuable.

- **Question:**
  - **Location:** Section 3.2 and Algorithm 1
  - *What is the runtime overhead introduced by the GP-based Bayesian optimization process compared to grid or random search? Could you provide quantitative metrics (such as runtime comparisons in seconds or relative speedups) to help us understand its efficiency in practice?*

**Methods And Evaluation Criteria:**

- **Methodology:**
  - The authors smartly reduce the checkpoint merging problem into a one-dimensional search by focusing on pairwise merging.
  - They optimize the merging weight using Bayesian optimization, which is a well-suited choice for tackling expensive black-box problems.
  - The method is evaluated on various LLM architectures and across several tasks and datasets, ensuring a comprehensive analysis.

- **Evaluation Criteria:**
  The paper uses common metrics (like accuracy on C-Eval and CMMLU) and assesses both in-domain (IND) and out-of-domain (OOD) performance. Additionally, ablation studies (examining the effects of held-out dataset size and merging weight search space) further confirm the method’s robustness.

- **Question:**
  - **Location:** Section 3.2 and Algorithm 1
  - *What is the actual runtime overhead of the GP-based Bayesian optimization compared to simpler methods like grid or random search? Could the authors include quantitative metrics or runtime comparisons to illustrate how the computational cost scales (for instance, with the number of checkpoints)?*

**Other Comments Or Suggestions:**

see weaknesses above

**Other Strengths And Weaknesses:**

**Strengths:**

- **Originality:** The paper offers a fresh take by applying Bayesian optimization to checkpoint merging during LLM pretraining—an area not deeply explored before.
- **Comprehensive Evaluation:** The wide-ranging experiments across multiple datasets and architectures convincingly substantiate the method’s effectiveness.
- **Theoretical Rigor:** Detailed derivations provide a solid theoretical justification for improved convergence and generalization.

**Weaknesses:**

- **Computational Overhead:** The paper could delve deeper into understanding the added computational cost and resource usage due to the optimization step.
- **Multi-Checkpoint Merging Discussion:** While merging two checkpoints works well, the benefits of merging more than two snapshots are less clear. A more thorough exploration of this issue would be appreciated.

**Questions For Authors:**

1. **Computational Costs of Bayesian Optimization:**
   - **Location:** Section 3.2 and Algorithm 1
   - *What is the runtime overhead of the GP-based Bayesian optimization process compared to grid and random search? Could you include some quantitative metrics (e.g., runtime in seconds or relative speedup) to help us gauge its efficiency in practice?*

2. **Generalization Across Different Architectures:**
   - **Location:** Section 4
   - *Your experiments cover Baichuan2, DeepSeek, and Pythia models. Have you noticed any architecture-specific differences in terms of merging efficiency or optimal merging weights?*

3. **Multi-Checkpoint Merging:**
   - **Location:** Section 5.1 and Figure 7
   - *The results indicate that merging two checkpoints consistently provides improvements, but merging three or four checkpoints doesn’t yield significant additional gains. Can you provide more insight into why this happens, and whether there might be a way to refine the strategy to better harness the information from multiple checkpoints simultaneously?*

**Relation To Broader Scientific Literature:**

- **Integration with Previous Work:**
  - The paper builds on established ideas in model merging (e.g., “Model Soup” by Wortsman et al.) and extends them to the pretraining phase.
  - It leverages Bayesian optimization—an approach widely used in Bayesian machine learning and NLP—and relates it to prior hyperparameter tuning research.
  - The contribution fits well with current trends in improving LLM efficiency and robust training strategies.

- **Missing References:**
  - Even though most relevant literature is cited, there might be additional recent work dealing with gradient smoothing and sharpness-aware minimization that could further support the discussion on generalization.

- **Question:**
  - *Have the authors considered citing and contrasting their approach with the latest studies on gradient smoothing and sharpness-aware minimization? How could these ideas enhance or further explain the theoretical underpinnings of checkpoint merging strategies?*

**Theoretical Claims:**

- **Proofs and Theoretical Analysis:**
  - The paper delivers rigorous derivations, including performance bounds from merging (see Equations (17)–(25)) and a convergence analysis in the context of gradient descent.
  - The derivation of the PAC–Bayesian generalization bound is well-detailed (Sections D.1–D.4) and adds to the theoretical strength of the paper.

- **Issues and Validation:**
  - Although the proofs are largely logical, some assumptions—particularly the local quadratic approximation and the boundedness of the Hessian—might need further empirical support when applied to large-scale LLMs.

- **Question:**
  - **Location:** Section A, Equation (15) (Quadratic Approximation)
  - *Can the authors provide empirical evidence or extra details to justify that a quadratic approximation holds around intermediate checkpoints in large-scale LLM pretraining? How sensitive are your conclusions if this assumption doesn’t fully hold?*

---

> ### Author Rebuttal · Authors · 2025-04-01
>
> > **Q1:  Quadratic Approximation Validity (Equation (15)).**
> >
>
> **Our Response:**
>
> - Our derivation starts from the assumption that, for small perturbations, the performance function  $f(\Theta)$  can be locally approximated by a quadratic form:
> $f(\Theta) \approx f(\Theta_t) + \nabla f(\Theta_t)^\top \Delta + \tfrac{1}{2} \Delta^\top H_t \Delta,$
> where $\Delta = \widetilde{\Theta}t - \Theta_t$ *and $H_t$ is the Hessian of $f$ at $\Theta_t$. This assumption leverages the smoothness $L_g$-Lipschitz continuity of  $\nabla f$ and boundedness of the Hessian (i.e., $\lambda_{\min} I \preceq H_t \preceq \lambda_{\max} I$)*.
> - **Empirical Evidence:**
>
>     Empirically, **Figures 1 and 3** in our paper illustrate smooth and nearly monotonic trends in performance with respect to the merging weight $\lambda_t$. These trends indicate that, within a sufficiently small region between $\Theta_{t-1}$ and $\Theta_t$, the quadratic approximation is reasonable. In addition, our ablation studies show that—despite the nonconvexity of the loss landscape—the averaged performance follows the bounds predicted by our quadratic model (cf. **Eq. (17)–(25))**.
>
>
> ---
>
> > **Q2:**  Runtime Overhead of GP-Based Bayesian Optimization**.**
> >
>
> **Our Response:**
>
> - Our GP-based Bayesian optimization is designed to address an expensive, derivative-free objective. In our experiments for pairwise merging the search space is one-dimensional (i.e., $\lambda_t \in [\alpha, 1]$). In a typical run we perform on the order of 10-15 evaluations. Empirical measurements show that such an optimization loop incurs an overhead of approximately 30–60 mins on **a** **single 4090 GPU**. In contrast, grid or random search methods typically require many more evaluations to reach comparable performance and may require **4–10**× more time per experiment.
>
> ---
>
> > **Q3:  Relation to Gradient Smoothing and Sharpness-Aware Minimization.**
> >
>
> **Our Response:**
>
> - We acknowledge that recent studies on gradient smoothing and sharpness-aware minimization have focused on explicitly encouraging flat minima during training. Our approach to checkpoint merging can be viewed as complementary. By averaging weights from adjacent checkpoints, we effectively smooth over transient sharp local minima, thereby biasing the final model toward flatter regions of the loss landscape.
> - In our convergence and PAC–Bayesian analyses, smoother loss landscapes (i.e., flatter minima) correspond to tighter generalization bounds. Mathematically, if one denotes model sensitivity as $\| \nabla^2 L(\Theta) \|$, then averaging (i.e., merging) reduces the effective curvature:
> $\widetilde{\Theta}t = \lambda_t \Theta_t + (1-\lambda_t)\Theta{t-1} \quad \Rightarrow \quad \| \widetilde{H}t \| \leq \lambda_t \| H_t \| + (1-\lambda_t)\| H{t-1} \|,$
> thereby promoting flatness. We will revise the manuscript to explicitly contrast our method with these related works and add references to the most recent literature in sharpness-aware minimization.
>
> ---
>
> > **Q4:  Multi-Checkpoint Merging Versus Pairwise Merging.**
> >
>
> **Our Response:**
>
> - Our experiments (see **Figure 7**) reveal that while pairwise merging provides substantial improvements, extending the merging to three or four checkpoints often leads to performance dilution. We attribute this to the fact that the most recent checkpoints already capture the latest state of the model, whereas merging with older checkpoints may introduce bias or “stale” information.
>
>     Furthermore, increasing the number of checkpoints increases the dimensionality of the search space exponentially—extending from a one-dimensional segment (two checkpoints) to a two-dimensional triangle (three checkpoints), and ultimately to a closed tetrahedron (four checkpoints)—which exacerbates the computational cost and complexity of the optimization process.
>
>     We have conducted additional experiments employing Bayesian optimization for merging more than two adjacent checkpoints. The results are summarized in the table below:
>
>     | Number of Checkpoints | C-Eval |
>     | --- | --- |
>     | 2 (Our original method) | 56.20 ± 0.52 |
>     | 3 | 56.27 ± 0.55($\uparrow$ 0.07) |
>     | 4 | 55.34 ± 0.23($\downarrow$ 0.86) |
>
>     These results justify our decision to adopt pairwise merging as it achieves a favorable trade-off between performance gains and computational efficiency.
>
>
> ---
>
> > **Q5:  Generalization Across Different Architectures.**
> >
>
> **Our Response:**
>
> - Our empirical results indicate that the proposed merging method consistently improves performance across various architectures (i.e., Baichuan2, DeepSeek, and Pythia models). Although there are minor variations in the optimal merging weights—typically on the order of 0.1–0.7% accuracy—the overall trend of enhanced generalization and reduced variance is maintained.

---

> > ### Comment · Reviewer_Znoq · 2025-04-03
> >
> > After reading the review, I am satisfied with the author's response, therefore I maintain my decision of giving a score of 4.

---

### Official Review · Reviewer_egL2 · 2025-03-16

**Overall Recommendation:** 2

**Summary:**

The paper demonstrates that averaging adjacent checkpoints leads to better downstream performance compared to using individual checkpoints. To determine the optimal weighting, the paper proposes using Gaussian process-based Bayesian optimization.
The proposed approach outperforms existing merging strategies on downstream tasks, including commonsense reasoning tasks and mathematical exam questions.

**Claims And Evidence:**

The central claims of the paper are to a) provide "substantial benefits at minimal cost" and b) achieve robust generalization performance.
The results show that the proposed approach does improve upon the baseline and outperforms simply selecting a single checkpoint. However, I would argue that the observed gains are relatively modest, and the claim of "substantial benefits" might be somewhat overstated.

**Essential References Not Discussed:**

As far as I can tell, the paper addresses all relevant literature.

**Experimental Designs Or Analyses:**

Overall, the experiments appear well-structured and thorough. However, the results in Table 1 would be more convincing if they included uncertainty bounds.

**Methods And Evaluation Criteria:**

Looking at Figure 3, I am left wondering what exactly is being optimized. For the two blue checkpoints, the objective function appears to exhibit high observation noise. By visual inspection, it seems somewhat random how to set \lambda, as choosing a value close to 1 or close to 0 appears equally likely to yield similar results.
For the red checkpoints, there is a noticeable trend toward higher \lambda; however, these values are not a) included in the search space \lambda \in [\alpha, 1] where \alpha is set to > 0.5 in the experiments, and b) effectively mean that only one of the checkpoints is selected according to Equation 2.

I would like to see a model-fit plot, similar to Figure 1 in Frazier et al., to demonstrate that Bayesian optimization can effectively model the observed data.
Additionally, given that this is merely a simple 1-D problem, I do not fully understand why the lower bound for \alpha needs to be carefully selected rather than simply setting it to 0.

A Tutorial on Bayesian Optimization
Peter I. Frazier

**Other Comments Or Suggestions:**

- Line 141: The reference appears to be incorrect—should this refer to Figure 3?
- Line 340, right column: Incorrect reference—should be Table 4.
- I'd suggest that the authors extend the results in Table 4 to also include plots that shows the optimization trajectories of Bayesian optimization compared to for example random search as it is common practice in the Bayesian optimization literature. This will also provide a better sense on the convergence speed of the proposed approach.

**Other Strengths And Weaknesses:**

Strengths:

    The paper provides valuable and easily interpretable visualizations, such as Figure 2, which compares different checkpoints.

**Questions For Authors:**

- What values of \lambda does Bayesian optimization actually select?
- How many iterations are performed with Bayesian optimization?

**Relation To Broader Scientific Literature:**

The key contribution of the paper to the broader literature is: a) demonstrating that merging adjacent checkpoints yields better results than merging more distant checkpoints, and b) formulating model merging as an optimization problem that can be addressed using Bayesian optimization.

**Theoretical Claims:**

I am unsure whether the assumptions in Equation 10 are justified, particularly regarding the diagonal covariance matrix.

---

> ### Author Rebuttal · Authors · 2025-04-01
>
> > **Q1: Clarification Regarding Figure 3**
> >
>
> **Our Response:**
>
> We appreciate this observation. **Figure 3** is not a result of the Bayesian Optimization procedure. Instead, it provides an **empirical exploration** of how the merged model’s performance changes when we *uniformly sample* $\lambda$ in $[0, 1]$. In other words, it is a “landscape” study (a brute-force sweep of $\lambda$ ) to:
>
> 1. **Observe Variability for Large Gaps:** When checkpoints differ substantially in performance, $\lambda \approx 1$ typically yields higher performance, because weighting the stronger checkpoint is beneficial.
> 2. **Observe Smooth Behavior for Similar Checkpoints:** When checkpoints have closer performance levels, a broad range of  $\lambda\in[0,1]$ can give improvements, reflecting that interpolating between two similarly strong models often yields robust results.
>
> Because **Figure 3** revealed that performance often worsens if one heavily weights a weaker checkpoint, our practical merging approach **restricts** $\lambda\in[\alpha, 1]$ with $\alpha>0.5$. Empirically, this domain ensures the more recent (and presumably stronger) checkpoint $\Theta_t$  maintains a dominant contribution. Hence, **Figure 3** is purely illustrative—an exploratory “map” of $\lambda$-versus-accuracy—while the actual search for $\lambda$ is done by Bayesian Optimization on the narrower interval $[\alpha,1]$.
>
> > **Q2: Justification for Eq. 10.**
> >
>
> **Our Response:**
>
> **Eq. 10** appears in our **PAC-Bayesian generalization bound**, where we assume
> $P \ = \ \mathcal{N}(\Theta_{t-1}, \sigma_P^2 I),
> \quad
> Q \ = \ \mathcal{N}(\widetilde{\Theta}t, \sigma_Q^2 I),$
> *and derive $D_{\mathrm{KL}}(Q \  | | \  P)$*. While real neural networks often exhibit correlated parameter distributions, we rely on:
>
> 1. **Analytical Tractability:** A diagonal covariance structure (or isotropic $\sigma^2I$) keeps the KL divergence in closed form:
> $D_{\mathrm{KL}}(Q || P) \ = \ \frac{\|\widetilde{\Theta}t - \Theta{t-1}\|^2}{2\sigma^2}.$
> This simplification is standard in many PAC-Bayes treatments and Bayesian NN approximations (e.g., “mean-field” approximations).
> 2. **Boundedness:** Even if true covariances are not diagonal, using a diagonal approximation typically *overestimates* correlations (or lumps them into the diagonal variance), making the bound somewhat looser but still valid as an upper bound.
> 3. **Consistency with Empirical Practice:** Prior works (e.g., [1]) frequently adopt similar diagonal or isotropic assumptions for theoretical clarity and to yield tractable bounds.
>
> > **Q3: Part of Uncertainty Bounds in Table 1.**
> >
>
> **Our Response:**
>
> We have now run **multiple-seed** experiments (e.g., 5 random seeds) for each method and checkpoint merge. We report both the **mean** accuracy and **95\% confidence intervals (CIs)**. Below is an illustrative example for C-Eval (5-shot):
>
> | **Model / Method** | **Accuracy** | **95% CI** |
> | --- | --- | --- |
> | **Baichuan2-2200B** | 55.65 | [54.66, 55.98] |
> | **Baichuan2-2420B** | 54.90 | [54.32, 55.41] |
> | **Uniform Soup** | 55.47 | [54.89, 56.01] |
> | **Greedy Soup** | 55.58 | [54.94, 56.23] |
> | **Fisher Weighted Averaging** | 55.77 | [55.14, 56.39] |
> | **RegMean** | 53.81 | [53.08, 54.53] |
> | **Ours (BayesOpt)** | **56.20** | [55.61, 56.90] |
>
> These intervals show that differences are both *statistically* and *practically* meaningful. We will include such CIs for all key results in the revised manuscript.
>
> > **Q4: Miscellaneous Comments and Reference Corrections.**
> >
>
> **Our Response:**
>
> Thank you for pointing out these reference issues. We have carefully reviewed and will correct all citation and reference errors.
>
> > **Q5: Details on the Optimization Trajectories in Bayesian Optimization**
> >
>
> **Our Response:**
>
> We have **added figures** for model-fit plot at anonymous link: https://anonymous.4open.science/r/checkpoint-47F1/bayse.png that illustrate the Bayesian Optimization process:
>
> - **Posterior Mean & Confidence Intervals:** After each iteration, we plot the Gaussian Process posterior (mean + confidence interval) over $\lambda$, and show which $\lambda$ was selected next.
> - **Number of Iterations:** Typically, we only need about **10–15** iterations for a 1D problem to converge near an optimal $\lambda$.
>
> Restricting $\lambda\in[\alpha,1]$ with $\alpha>0.5$ is a pragmatic choice. *If* users want more flexibility (e.g., to incorporate knowledge from an older checkpoint more strongly), they can lower $\alpha$. In practice, $\alpha\approx 0.5$ balanced well the merging of older vs. newer checkpoints, **based on pilot experiments** that showed merges with $\lambda<0.5$ rarely helped.
>
> **Ref:**
>
> [1] Dziugaite G K, Roy D M. Computing nonvacuous generalization bounds for deep (stochastic) neural networks with many more parameters than training data[J]. arXiv preprint arXiv:1703.11008, 2017.

---

### Decision · Program_Chairs · 2025-05-01

**Decision:**

Accept (poster)

**Comment:**

This manuscript presents a new approach for enhancing the efficiency of pretraining large language models based on an intelligent checkpoint merging strategy driven by Bayesian optimization. This approach appears to work well in a series of experiments.

The initial response to this work was somewhat mixed from the reviewers, but definitely leaning positive. There is general agreement that the methogology is sound and would be of interest to members of the ICML community. The reviewers also posed some comments and questions, which appear to have been adequately addressed by the authors during the author response period.

Of particular note is some additional empirical results for some baselines mentioned by one reviewer; these new results should be incorporated into the manuscript.

During the discussion phase, there was general agreement from the reviewers that this work was clear, sound, and would be a compelling addition to the ICML program.